# Quantum confined peptide assemblies with tunable visible to near-infrared spectral range

Kai Tao [1], Zhen Fan[2,3,4], Leming Sun[5], Pandeeswar Makam[1], Zhen Tian[6], Mark Ruegsegger[2], Shira Shaham-Niv[1], Derek Hansford[2], Ruth Aizen[1], Zui Pan [7], Scott Galster[8], Jianjie Ma[3,9], Fan Yuan[10], Mingsu Si[11], Songnan Qu[6], Mingjun Zhang[2,3,12], Ehud Gazit[1,13] & Junbai Li [14]

Quantum confined materials have been extensively studied for photoluminescent applications. Due to intrinsic limitations of low biocompatibility and challenging modulation, the utilization of conventional inorganic quantum confined photoluminescent materials in bio-imaging and bio-machine interface faces critical restrictions. Here, we present aromatic cyclo-dipeptides that dimerize into quantum dots, which serve as building blocks to further self-assemble into quantum confined supramolecular structures with diverse morphologies and photoluminescence properties. Especially, the emission can be tuned from the visible region to the near-infrared region (420 nm to 820 nm) by modulating the self-assembly process. Moreover, no obvious cytotoxic effect is observed for these nanostructures, and their utilization for in vivo imaging and as phosphors for light-emitting diodes is demonstrated. The data reveal that the morphologies and optical properties of the aromatic cyclo-dipeptide self-assemblies can be tuned, making them potential candidates for supramolecular quantum confined materials providing biocompatible alternatives for broad biomedical and opto-electric applications.

[1] Department of Molecular Microbiology and Biotechnology, George S. Wise Faculty of Life Sciences, Tel Aviv University, 6997801 Tel Aviv, Israel. [2] Department of Biomedical Engineering, College of Engineering, The Ohio State University, Columbus, OH 43210, USA. [3] Dorothy M. Davis Heart and Lung Research Institute, The Ohio State University Wexner Medical Center, Columbus, OH 43210, USA. [4] Department of Polymeric Materials, School of Materials Science and Engineering, Tongji University, 201804 Shanghai, China. [5] School of Life Sciences, Northwestern Polytechnical University, 710065 Xi'an, China. [6] State Key Laboratory of Luminescence and Applications, Changchun Institute of Optics, Fine Mechanics and Physics, Chinese Academy of Sciences, 130033 Changchun, China. [7] College of Nursing and Health Innovation, University of Texas at Arlington, Arlington, TX 76019, USA. [8] Rockefeller Neuroscience Institute and Department of Neuroscience, WVU School of Medicine, 9100 Morgantown, West Virginia, USA. [9] Department of Surgery, The Ohio State University, Columbus 43210, USA. [10] Department of Biomedical Engineering, Duke University, Durham, NC 27708, USA. [11] Key Laboratory for Magnetism and Magnetic Materials of MOE, Key Laboratory of Special Function Materials and Structure Design, Ministry of Education, Lanzhou University, 730000 Lanzhou, China. [12] Nurological Institute and Department of Surgery, The Ohio State University, Columbus, OH 43210, USA. [13] Department of Materials Science and Engineering, Iby and Aladar Fleischman Faculty of Engineering, Tel Aviv University, 6997801 Tel Aviv, Israel. [14] Beijing National Laboratory for Molecular Sciences, CAS Key Lab of Colloid, Interface and Chemical Thermodynamics, Institute of Chemistry, Chinese Academy of Sciences, 100190 Beijing, China. These authors contributed equally: Kai Tao, Zhen Fan  Correspondence and requests for materials should be addressed to M.Z. (email: zhang.4882@osu.edu) or to E.G. (email: ehudg@post.tau.ac.il) or to J.L. (email: jbli@iccas.ac.cn)

Quantum confined (QC) materials, such as quantum dots (QDs), have been widely employed for imaging due to their remarkable photoluminescent properties[1–4]. However, the state-of-the-art inorganic QC constituents, such as cadmium-based QDs, are intrinsically cytotoxic, thus extremely limiting their applications[5,6]. Although organic fluorescent dyes allow to overcome the potential cytotoxicity to some extent, several concerns, including weak sustainability and photobleaching, narrow color spectrum, and in some cases complicated synthesis procedures, still practically impede their utilization[4,7]. The quest for eco-friendly, organic, tunable, and flexible QC alternatives with improved and stable photoluminescence is continuously ongoing[8–10].

Accumulating studies demonstrate that aromatic linear-dipeptides, with the representative model of diphenylalanine (FF)[11], can self-assemble into nanostructures with remarkable physiochemical features[12,13], such as optic[14,15], electrical[16], ferroelectric[17] (including piezoelectric[18] and pyroelectric[19]) properties[20–23]. Especially, the supramolecular morphologies and properties can be easily modified by amino acids substitutions[24], covalent conjugation[25] or co-assembly with external moieties[26,27]. For example, upon substitution of one F with tryptophan (W), self-assembling FW nanostructures present a smaller bandgap of 2.25 eV, compared to 3.25 eV of FF nanotubes[28], thus showing improved conductive[29] and photoluminescent properties[24]. Moreover, recent studies revealed that cyclo-dipeptides with backbones of 2,5-diketopiperazine configurations, derived from dehydration condensation of linear dipeptides[16,30], self-assemble into photoluminescent nanostructures different from their linear counterparts[31,32].

Here, we demonstrate the modulation of W-based aromatic cyclo-dipeptides self-assemblies into QC microstructures with photoluminescent properties. In particular, the emissions are tuned throughout the visible and near-infrared region by modifying the self-assembly process. These wide spectral range QC structures can thus be used as bio-inspired phosphors for the fabrication of light-emitting diodes (LEDs) and for in vivo bioimaging without any obvious cytotoxicity. The results reveal that the morphologies and optical properties of the W-based aromatic cyclo-dipeptide self-assemblies can be tuned to achieve broad fluorescence emissions, making them promising candidates as supramolecular QC materials for biomedical and opto-electric applications.

## Results

**Dimeric QDs as building blocks for self-assembly**. Aiming to examine their potential as QC materials, we characterized the photoluminescent properties of two tryptophan (W)-containing aromatic cyclo-dipeptides[33], cyclo-phenylalanine-tryptophan (cyclo-FW), and cyclo-WW (Fig. 1a), both dissolved in methanol (MeOH). Fluorescent characterization demonstrated red shifts of the molecular excitation to 305 nm, compared to 285 nm for the monomers (Fig. 1b), indicating that the cyclo-dipeptides indeed self-assembled, which was further confirmed by dynamic light scattering (DLS) detections (Supplementary Fig. 1). UV-Vis absorption spectra revealed that the cyclo-dipeptides had spike-like absorbance, showing three peaks at 273 nm, 280 nm, and 289 nm (Fig. 1c), characteristic of the formation of QD structures[34,35]. The diameter of the QDs was calculated to be ~2.24 nm (for calculation details, see Methods), around two-fold of the dimension of the monomer. In addition, mass spectrometry (MS) analysis detected the dimeric molecular weights (MW), along with the corresponding monomeric molecular mass (Fig. 1d), indicating the presence of dimers. Therefore, it can be concluded that the cyclo-dipeptide monomers first formed dimers, which

behaved as QDs and served as the fundamental building blocks to further organize into supramolecular structures (Fig. 1e). Theoretical calculations using density functional theory demonstrated that the spatial distributions of the highest occupied and lowest unoccupied molecular orbitals of the dipeptides were mainly concentrated on the side-chain indole rings (Fig. 1f). Specifically, the band gaps ($\Delta E$) of cyclo-FW and cyclo-WW were calculated to be 3.63 eV and 3.56 eV, respectively, indicating their wide-gap semiconductive nature[36,37]. This implies that the dimerization was mostly driven by $\pi$–$\pi$ interactions between aromatic side-chains, especially the indole rings. Particularly, the aromatic interactions could induce the through-space conjugation of the electron clouds from adjacent indole rings, thus restricting the molecular motions[38,39] and underlying the molecular basis for QC regions[34]. The QC effects and restricted molecular activities resulted in the release of excitation energy exclusively as emitted light[10,12].

**Modulation of the self-assemblies morphology and visible fluorescence**. The orderly organized QDs inside the supramolecular self-assemblies result in extensive QC effects along with intrinsic photoluminescent properties[12]. When excited at 370 nm, the cyclo-dipeptides solutions displayed fluorescence in the visible region, with emission at 460 nm for cyclo-FW and at 425 nm, accompanied by a smaller peak at 520 nm, for cyclo-WW (Fig. 2a). Correspondingly, the solutions showed blue-green color under UV light (365 nm) (Fig. 2a, insets). Scanning electron microscopy (SEM) revealed that the two cyclo-dipeptides self-assembled into distinct supramolecular structures, as needle-like crystals were formed by cyclo-FW (Fig. 3a), while cyclo-WW assembled into spherical nanoparticles (Fig. 3b). The maximal emission of the cyclo-FW assemblies demonstrated a red-shift as excitation wavelength was increased (Supplementary Fig. 2a), indicating that heterogeneous superstructures of different sizes, arising from the dynamic self-organization, co-existed in the solution[40,41].

The ease of complexation of indole rings suggests that these self-assemblies could be doped by coordination with metal ions[42]. Previous studies revealed that Zn(II) can participate in amphiphilic peptides self-assembly[24,43,44] and interferes with amyloid proteins aggregation[45,46]. Indeed, by introducing Zn(II), the emission of cyclo-WW assemblies (cyclo-WW + Zn(II)) was clearly enhanced, showing a narrow peak at 520 nm with a full width at half maximum of only 18 nm (Fig. 2b), leading to a luminous green color under UV light and a quantum yield (QY) of 16% (Fig. 2b, inset). Intriguingly, atomic force microscopy (AFM) experiment indicated the presence of only small nanoparticles, ~3.0 nm in diameter (Fig. 3c; Supplementary Fig. 3), approximately the dimension of a dimer. Correspondingly, DLS analysis showed that the size of the structures was ~2.88 nm, with no larger particles present (Supplementary Fig. 4). The uniform size distribution resulted in a consistent maximal emission, regardless of excitation wavelength (Supplementary Fig. 2b). These findings demonstrated that following the coordination with Zn(II), the cyclo-WW self-assembly halted at the dimerization stage.

Another strategy to modulate the supramolecular photoluminescence is to substitute the constituents with their enantiomers[47]. When W was replaced with its D-type enantiomer in cyclo-FW, hereby designated cyclo-Fw, the fluorescent emission shifted to 430 nm (Fig. 2c). SEM and DLS analyses revealed that cyclo-Fw self-assembled into multi-branched nano-flower architectures (Fig. 3d; Supplementary Figs. 5a, 6a), showing red-shifted maximal emission upon various excitation wavelengths (Supplementary Fig. 2c).

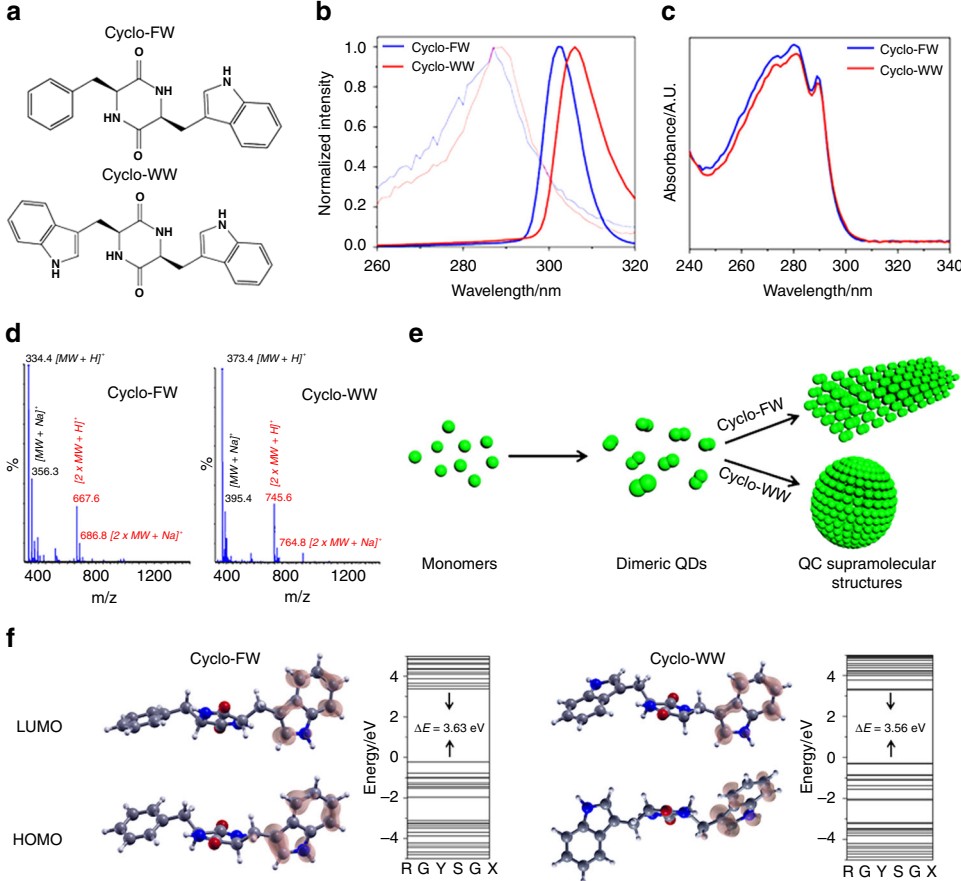

**Fig. 1** Cyclo-dipeptide dimers serve as QDs to self-assemble into QC nanostructures. **a** Molecular structures of cyclo-FW and cyclo-WW. **b** Excitation spectra of cyclo-dipeptide monomers (thin light curves, 0.05 mM) and self-assemblies (thick dark curves, 5.0 mM) in MeOH. The excitation wavelengths red shifted from 285 nm to ~305 nm after self-assembly. **c** UV-Vis absorption spectra of the cyclo-dipeptides after self-assembly, showing spike-like absorptions at 273 nm, 280 nm, and 289 nm, characteristic of the formation of QD structures. **d** MS spectra of cyclo-FW and cyclo-WW at 5.0 mM in MeOH. The dimeric MW is marked in red, and the monomeric MW is marked in black. **e** Schematic representing the process of cyclo-dipeptides self-assembly: the monomers form dimeric QDs, which serve as the building blocks to self-assemble into larger QC architectures. **f** Calculated molecular orbital amplitude plots and energy levels of the highest occupied and lowest unoccupied molecular orbitals of cyclo-FW and cyclo-WW, showing band gaps of 3.63 eV and 3.56 eV, respectively

Furthermore, the well-established reductive property of W[48] could be utilized to modulate the fluorescence of the cyclo-dipeptides self-assemblies. When introducing Cu(II), a weak oxidant[49], into cyclo-WW assemblies (cyclo-WW + Cu(II)), a fluorescent emission appeared at 465 nm, with a QY of 8% (Fig. 2d). A similar emission was also observed after irradiation with UV light (365 nm) (cyclo-WW + UV) due to the UV-induced radical oxidation (Fig. 2e), implying a potential for photo-stimulated applications. HPLC analysis confirmed that the conversion was complete and the oxidized cyclo-WW was pure (Supplementary Fig. 7). Fluorescent decay experiments demonstrated that after oxidation, the lifetime increased (6.3 ns for cyclo-WW + Cu(II), 8.0 ns for cyclo-WW + UV) compared to cyclo-WW (5.6 ns) (Fig. 2f; Supplementary Fig. 8a), confirming that the redox reactions indeed took place. SEM and DLS characterizations showed that the oxidized cyclo-WW self-assembled into spherical nanoparticles, several hundred nanometers to micrometers in diameter (Fig. 3e, f; Supplementary Figs. 5b, c). In both cases, the maximal emission red-shifted (Supplementary Figs. 2d, e), demonstrating that the spherical nanoparticles further aggregated to form larger particles (Supplementary Figs. 6b, c). In fact, massive precipitates could be found at the bottom of the cyclo-WW + UV and cyclo-WW + Cu(II) solutions after one week and one month, respectively.

**Mechanisms underlying the fluorescence modulations.** Understanding the mechanisms underlying the modulation of the fluorescent properties of the self-assemblies is critical for their future applications[50]. A absorption peak at 515 nm, corresponding to ligand (peptide)-to-metal charge transfer[51], emerged upon mixing cyclo-WW and Zn(II) in solution (Fig. 4a), indicating the formation of coordinated architectures. Notably, this absorption peak resulted in a color change from the original white/light yellow of cyclo-WW to pink (Supplementary Fig. 9). In contrast, no new band appeared when Zn(II) was introduced into a monomeric cyclo-WW solution (Supplementary Fig. 10), confirming that the dimers were indeed the form that complexed with Zn(II). Especially, the charge transfer could deliver the excited electrons, resulting in reduced fluorescence decay time, from 5.6 ns of cyclo-WW to 3.6 ns (Fig. 2f; Supplementary Fig. 8a). To determine the stoichiometric ratio of cyclo-WW and Zn(II), a Job Plot analysis was performed, showing the intersection point at a cyclo-WW proportion of ~0.7 (Fig. 4b). This result indicates that the stoichiometry of cyclo-WW dimers and Zn(II) was approximately 1:1 (0.35:0.3). The [1]H-NMR spectra revealed that after coordination, the hydrogen atoms in the backbone showed downfield shifting (Fig. 4c; Supplementary Fig. 11), indicating that the shielding effect became weaker. This suggests that the diketopiperazine ring contributed to the coordination by

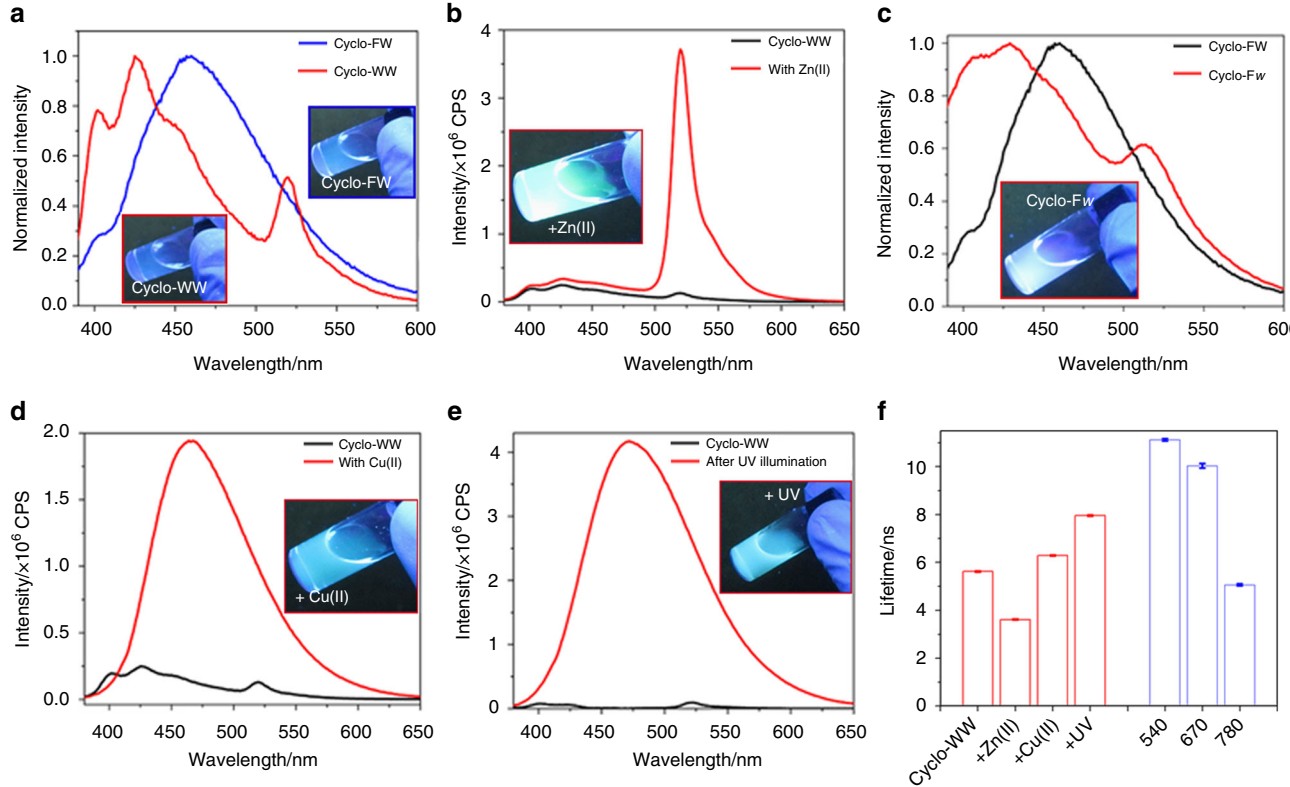

**Fig. 2** Visible fluorescence properties and modulation of the self-assemblies. **a–e** Fluorescent emission of: (**a**) cyclo-FW (blue) and cyclo-WW (red) self-assemblies, (**b**) cyclo-WW + Zn(II), (**c**) cyclo-Fw, (**d**) cyclo-WW + Cu(II), (**e**) cyclo-WW + UV. The insets show the corresponding solutions under UV light (365 nm). In **b–e**, intrinsic fluorescent emissions of the corresponding cyclo-dipeptides were added and marked in black for comparison. **f** Lifetime statistics of cyclo-WW self-assemblies. Left panel (red): cyclo-WW in MeOH (cyclo-WW: 370/425; + Zn(II): 370/520; + Cu(II): 370/465; + UV: 370/465). Right panel (blue): cyclo-WW + Zn(II) in DMSO (540/610; 670/712; 740/817). Error bars on lifetime measures show standard deviations for three replicates

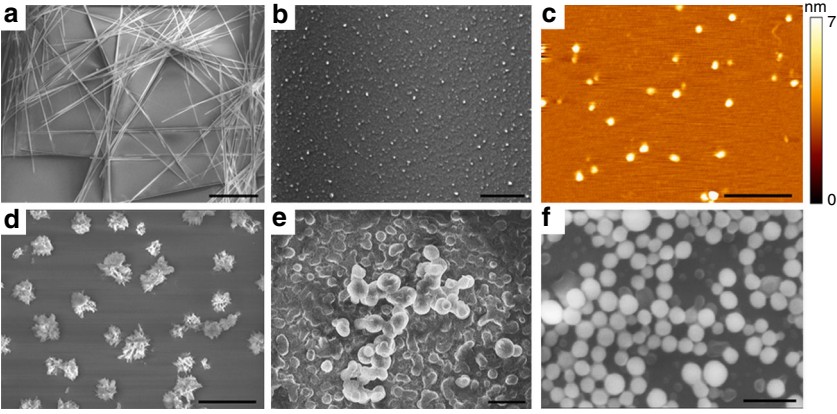

**Fig. 3** SEM and AFM images of the cyclo-dipeptides QC self-assemblies in MeOH. **a** Needle-like cyclo-FW crystals., Scale bar: 40 μm. **b** Spherical cyclo-WW nanoparticles, Scale bar: 20 μm. **c** Dimeric QDs of cyclo-WW + Zn(II), Scale bar: 500 nm. **d** Nano-flower architectures of cyclo-Fw, Scale bar: 20 μm. **e** and **f** Larger spherical nanoparticles of (**e**) cyclo-WW + Cu(II) and (**f**) cyclo-WW + UV. Scale bar: 2 μm

supplying electrons to interact with Zn(II)[52]. In contrast, the hydrogen atoms in the indole rings showed upfield shifting (Fig. 4c; Supplementary Fig. 11), indicating that the delocalization of the π-electrons on the aromatic rings became stronger[52]. This demonstrates that the indole rings formed aromatic interactions with each other and through-space conjugation of electrons took place, consistent with the theoretical calculations. Fourier-transform infrared spectroscopy (FTIR) characterizations demonstrated that in the presence of Zn(II), the N–H stretching

vibration in diketopiperazine rings red-shifted from the original 3200 cm$^{-1}$ to 3071 cm$^{-1}$ [53] due to the decrease of the bond energies resulting from the metal ion adsorption (Fig. 4d, peak 1) [54], indicating that the nitrogen atoms in the backbone diketopiperazine rings contributed to the complexation with Zn(II). Combined with the NMR analysis, a plausible molecular mechanism of cyclo-WW coordination with Zn(II) is depicted in Fig. 4e. The Zn(II) ion is embedded in a dimer of cyclo-WW, coordinating with two diketopiperazine rings, while the

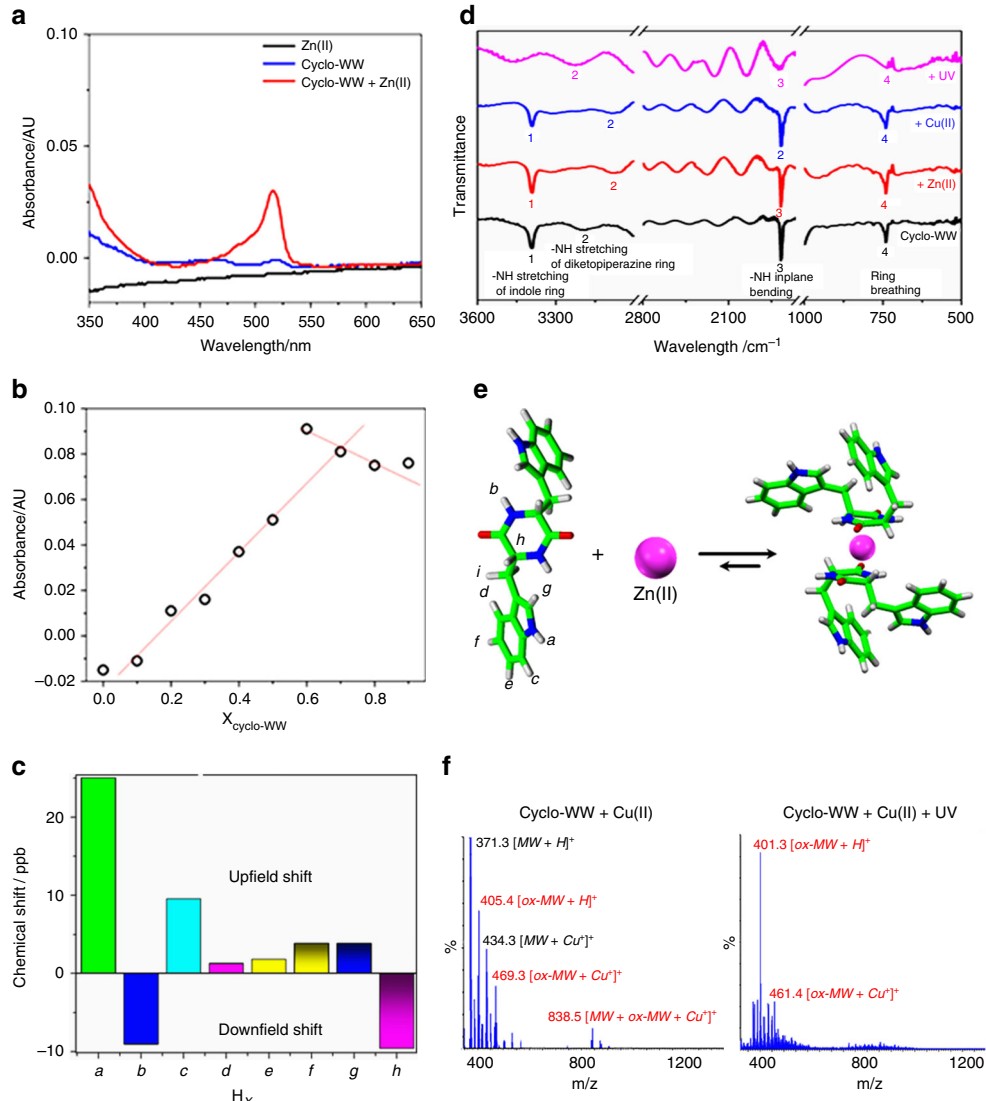

**Fig. 4** Mechanisms underlying fluorescence modulation. **a** UV-Vis absorption spectra of cyclo-WW self-assemblies in the absence or presence of Zn(II) in MeOH. A peak at 515 nm appeared when complexing cyclo-WW and Zn(II). **b** Job Plot analysis of cyclo-WW with Zn(II) at different ratios, with the total molar concentration fixed at 15.0 mM. The red lines were added for guideline, showing the intersection point at cyclo-WW proportion of 0.7. **c** Chemical shifts of cyclo-WW hydrogen atoms upon coordination with Zn(II), compared to the peptide alone. The hydrogen atoms in different chemical environments are marked with italicized letters and labeled in **e**. **d** FTIR spectra of cyclo-WW self-assemblies in the absence of metal ion/UV irradiation. The peaks of the active bonds are numerically marked. Note that the IR spectra were vertically moved for clarity. **e** Schematic presentation showing the possible molecular mechanism of cyclo-WW dimer coordination with Zn(II): the backbone diketopiperazine rings contribute to the complexation through nitrogen atoms, while the side-chain indole rings form aromatic interactions. **f** MS spectra of cyclo-WW + Cu(II) and cyclo-WW + Cu(II) + UV, showing the MW of oxidized cyclo-WW (marked in red) and reduced Cu(I), confirming the redox reactions in the solutions

side-chain indole rings form $\pi$–$\pi$ interactions. It is possible that the complexation induces hindrance against further aggregation of the dimers, thus resulting in the dimers separated from each other and showing stable photoluminescence regardless of the excitation wavelength.

The larger size and aggregating nature of the cyclo-WW + Cu (II) and cyclo-WW + UV nanospheres resulted in time-resolved parabolic evolution of the emission intensities (Supplementary Figs. 12, 13). In fact, the weak oxidation of Cu(II) induced slower aggregation dynamics, thus leading to a continuous increase of the fluorescent emission at 465 nm, with an excitation of 370 nm, persisting over one month (Supplementary Fig. 13a, black circles). This also resulted in larger aggregations with a long-wavelength 520 nm emission, following excitation at 395 nm, constantly increasing over this period (Supplementary Fig. 13a, red circles).

As a control, the emission of cyclo-WW + Zn(II) decayed due to quenching (Supplementary Fig. 14). The redox reaction was verified by MS analysis, showing an $m/z$ corresponding to the MW of oxidized cyclo-WW in the cyclo-WW + Cu(II) solution, in addition to the dominant peak of native cyclo-WW (Fig. 4f, left panel). When combining both Cu(II) oxidation and UV irradiation (cyclo-WW + Cu(II) + UV), the synergistic effect dramatically accelerated the oxidation reaction, leading to an MS profile predominantly comprised of the $m/z$ of oxidized cyclo-WW (Fig. 4f, right panel). Additionally, in both mass-spectra, the MW of Cu(I)-conjugated peptides was detected, indicating that the Cu(II) ions have been reduced. In contrast, in cyclo-WW + Zn(II), a chloride ion was always integrated, thus counteracting the positive charge of Zn(II) (Supplementary Fig. 15).

The oxidation mechanism of Cu(II) was further verified using other metal ions with higher oxidative capability. When replacing Cu(II) with Ag(I) or [AuCl$_4$](-I), the sample solutions showed the same emission spectra with a maximum at 465 nm and blue-green color under UV light (Supplementary Fig. 16a). Notably, the intense redox reduced the metal ions to elementary metals (Supplementary Fig. 16b). The different modulation mechanisms of Zn(II) and Cu(II) could also be confirmed by mixing with cyclo-FW, which did not complex with Zn(II) but showed enhanced fluorescence at 465 nm with a QY of 12% in the presence of Cu(II) (Supplementary Fig. 17).

The similar emission spectra of cyclo-WW + Cu(II) and cyclo-WW + UV suggested that the oxidized cyclo-WW did not complex with metal ions. In fact, cyclo-WW + Zn(II) + UV had the same fluorescence emission as cyclo-WW + UV and cyclo-WW + Cu(II) + UV (Supplementary Fig. 18), rather than that of cyclo-WW + Zn(II) (Fig. 2b). FTIR analysis demonstrated that in cyclo-WW + Cu(II) and cyclo-WW + UV, the vibration of N–H stretching of indole rings (3393 cm$^{-1}$) significantly attenuated (Fig. 4d, peak 1)[53], indicating that the redox took place at the N–H bonds of the side-chains. Combined with the MS data, we propose that the two nitrogen atoms of the indole rings were conjugated with oxygen to form –N = O bonds. In addition, the intensity of the ring breathing vibration (742 cm$^{-1}$) also declined (Fig. 4d, peak 4)[53], suggesting that the redox severely disrupted the conformation of the rings[54]. It is speculated that the oxidation changed the electronic density and steric structures of the indole rings, thereby hindering the interactions of oxidized cyclo-WW with metal ions.

**Near-infrared (NIR) fluorescence properties of the self-assemblies.** A typical characteristic of QC materials is the dependence of their emission colors on particles size[55]. The excitation-dependent photoluminescent feature of the self-assemblies suggests that their emissions might be red-shifted to even longer wavelengths by introducing inhomogeneous size distributions[40], for instance by using solvents that can facilitate the self-assembly. Indeed, when replacing MeOH with the more polar DMSO as a solvent for cyclo-WW + Zn(II), the extensive aggregation resulted in molecular excitation red-shift, from 305 nm in MeOH to 310 nm (Supplementary Fig. 19), and the dimers self-assembled into larger spherical nanoparticles, 63.6 ± 12.2 nm in diameter (Fig. 5a; Supplementary Figs. 20, 21). X-ray diffraction analysis revealed distinct sharp peaks with high intensities (Supplementary Fig. 22), indicating the high crystallinity of the nanospheres, consistent with the crystallized nature of the QDs. These results illustrated the well-organized, periodical nano-lattice arrangement within the assemblies, thus confirming the directional organization of the dimeric QDs and the extensive internal QC effects.

Fluorescent characterization of the cyclo-WW + Zn(II) nanospheres in DMSO demonstrated both visible and NIR fluorescence under a wide range of excitation wavelengths (Fig. 5b). Specifically, the peaks at 615 nm, 712 nm, and 817 nm were emitted using excitation wavelengths between 520 and 780 nm (Fig. 5c; Supplementary Fig. 23), with lifetimes of 11.1 ns, 10.0 ns, and 5.1 ns, respectively (Fig. 2f; Supplementary Fig. 8b), suggesting that as aggregation progressed, the excited electrons became more stable. This indicates a promising potential of the supramolecular structures to be used as bio-inspired alternatives for stable, long-term imaging. Correspondingly, the solution displayed distinct colors under different excitation wavelengths (Fig. 5d). The QYs were measured to be 18% and 27% for emissions at 712 nm and 817 nm, respectively. Photobleaching evaluation experiments demonstrated that the photoluminescence of the nanospheres remained stable after continuous

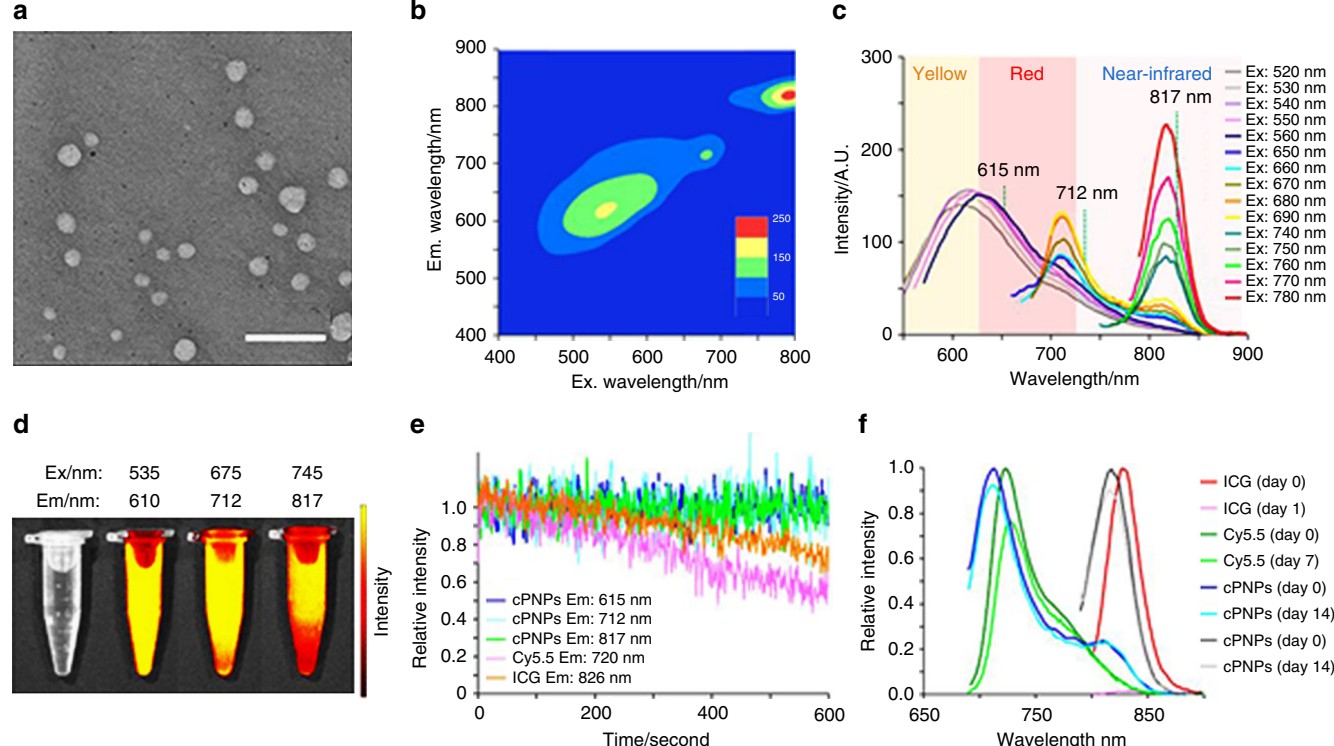

**Fig. 5** NIR fluorescence properties of cyclo-WW + Zn(II) in DMSO. **a** TEM image of the self-assembled nanospheres. Scale bar: 300 nm. **b** Emission vs. excitation profile of the nanospheres. **c** Extracted emission spectra of the nanospheres. **d** Visible to NIR photos of cyclo-WW + Zn(II) DMSO solution under various wavelengths. **e** Photobleaching and (**f**) photostability characterization of the cyclo-dipeptide nanoparticles (abbreviated as cPNPs) and of the fluorescent dyes ICG and Cy5.5, as controls

irradiation for 600 s, compared to the significant fluorescence decay observed for organic fluorescent dyes (indocyanine green (ICG) and cyanine 5.5 (Cy5.5)) (Fig. 5e). In addition, time-resolved characterization showed the fluorescence of the nanospheres to be stable even after exposure to natural light for two weeks, indicating their high photostability. In contrast, ICG lost all fluorescence after 1 day and Cy5.5 lost 20% of the fluorescence after 1 week (Fig. 5f).

**Application of the QC fluorescent self-assemblies**. The photoluminescent nature endows the peptide QC self-assemblies the ability to be used for photo-stimulated devices, such as LEDs. By applying a mixture of dried cyclo-WW + Zn(II) dots in MeOH and polydimethylsiloxane (PDMS) onto an indium gallium nitride (InGaN) chip, a prototypical LED device using peptide self-assemblies as phosphors was fabricated (Fig. 6a). When applying voltages, bright green light was illuminated, as shown in the inset of Fig. 6a. Spectroscopic investigations demonstrated an emission around 550 nm regardless of the excitation wavelength (Fig. 6b), thus showing remarkable emission specificity. The red-shift of 30 nm from 520 nm (Fig. 2b) was probably due to aggregation that occurred during MeOH evaporation.

Furthermore, the bioinspired nature and the notable emission up to the NIR region suggest the potential utilization of cyclo-WW + Zn(II) nanospheres in biological systems. Based on an in vitro cytotoxicity analysis, the peptide nanoparticles showed good biocompatibility towards B16BL6 (murine melanoma cell line), MCF-7 (human breast cancer cell line), and HaCaT (human skin cell line) cells (Fig. 6c). Further, subcutaneous injection of the nanospheres into nude mice followed by NIR fluorescence imaging revealed distinct visible and NIR fluorescent signals at the injection site (Fig. 6d). Notably, the fluorescent signals were stable, showing no decay for one week, thus highlighting the possibility of utilizing the photoluminescent QC assemblies for in vivo bio-imaging applications. In addition, the advantage of easy modifications, such as specific targeting and controllable assembly/dis-assembly, facilitates simple functionalization of the assemblies[56], thus exemplifying their promising potential for targeted therapy and controllable drug release.

**Influence of aromatic side-chains**. The important roles aromatic side-chains play during self-assembly suggest that the supramolecular morphologies and photoluminescent properties can be modulated by changing the aromatic moieties[57]. Therefore, three cyclo-dipeptides comprised of different aromatic amino acids, i.e., cyclo-dihistidine (cyclo-HH), cyclo-diphenylalanine (cyclo-FF), and cyclo-dityrosine (cyclo-YY) (Fig. 7a), were examined under the same conditions. Morphological characterization showed that the cyclo-dipeptides self-assembled into diverse supramolecular structures. Specifically, cyclo-HH formed nanofibers in MeOH (Fig. 7b), cyclo-FF formed spherical nanoparticles in DMSO (Fig. 7c), and cyclo-YY assembled into nanorods in DMSO (Fig. 7d). AFM analysis demonstrated small aggregates less than 4 nm in height at a lower concentration (0.5 mM), with

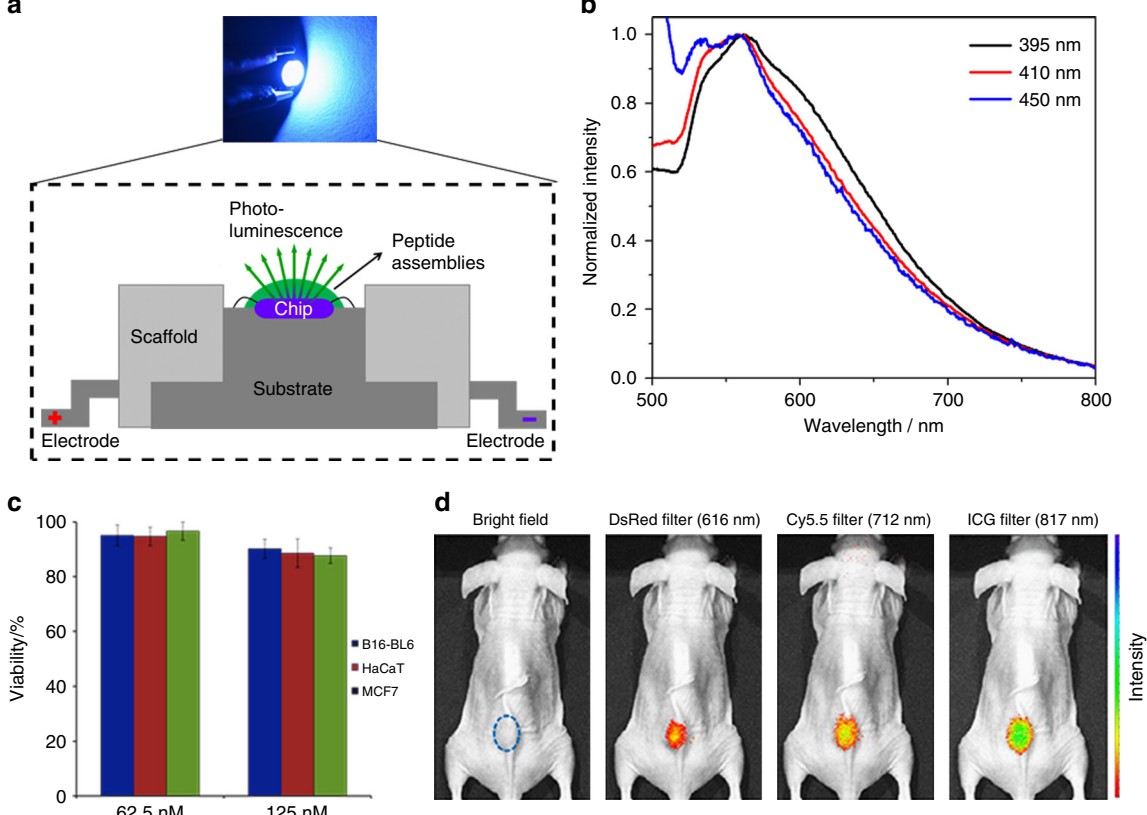

**Fig. 6** Application of the cyclo-WW + Zn(II) QC self-assemblies. **a** Schematic presentation of the LED setup using dried cyclo-WW + Zn(II) assembled in MeOH as phosphors. The upper inset shows the working depiction of a prototype, emitting bright green light (ex: 450 nm). **b** Spectroscopic characterization of the LED photoluminescence using three excitation wavelengths, as indicated, showing the same emission at 550 nm. **c** Cytotoxicity test of the cyclo-WW + Zn(II) nanospheres in DMSO (62.5 nM and 125 nM) towards B16-BL6, HaCaT, and MCF7 cells. Viability relative to untreated controls ± sd, designated as error bars, is shown based on three repeats and averaged. **d** In vivo whole body NIR fluorescent imaging following subcutaneous injection of the nanospheres (50 μL, 2.7 mM) into nude mice, showing notable emissions under various excitations. The dotted circle indicates the location of injection

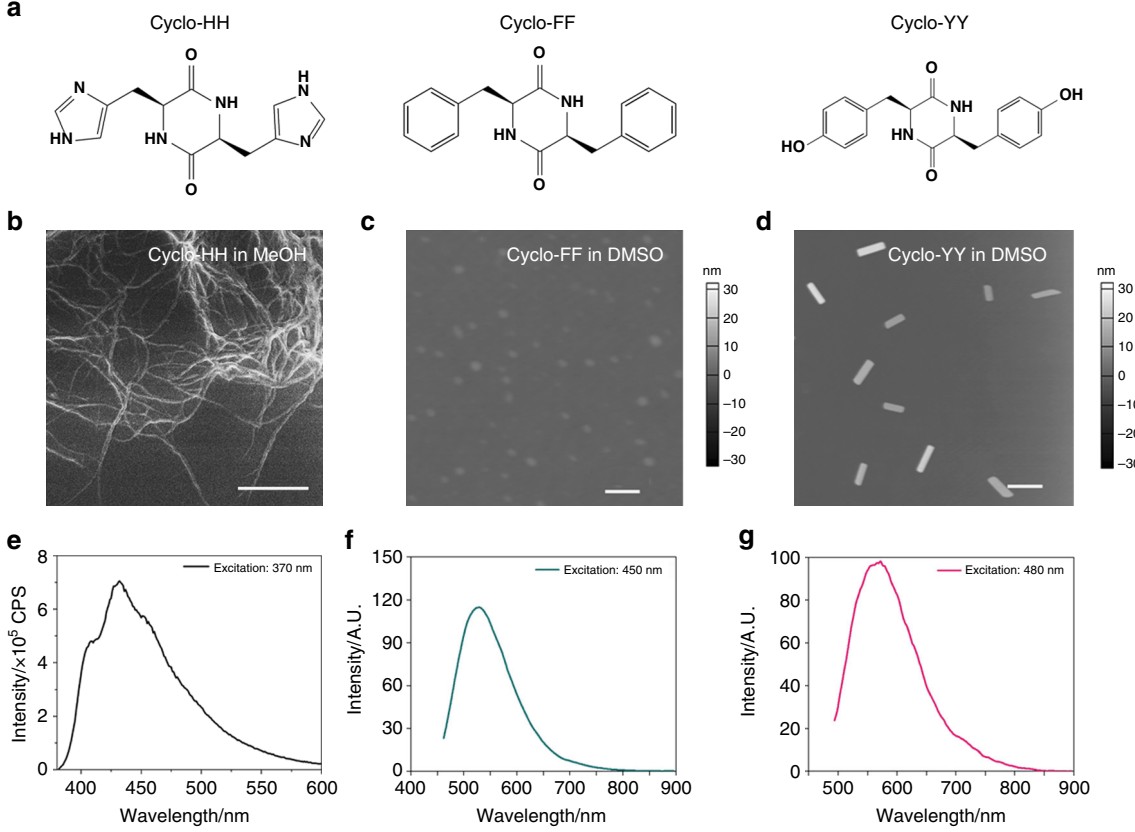

**Fig. 7** The effect of diverse aromatic side-chains. **a** Molecular structures of cyclo-HH, cyclo-FF, and cyclo-YY. **b–d** Microscopic images showing the self-assemblies of (**b**) cyclo-HH in MeOH (scale bar: 1 μm), (**c**) cyclo-FF in DMSO (scale bar: 50 nm), and (**d**) cyclo-YY in DMSO (scale bar: 500 nm). **e–g** Fluorescent emission of (**e**) cyclo-HH in MeOH, (**f**) cyclo-FF in DMSO, and (**g**) cyclo-YY in DMSO

dots for cyclo-HH, cyclo-FF, and thin nanofibers for cyclo-YY (Supplementary Fig. 24). This suggested that like cyclo-FW and cyclo-WW, the self-assemblies of these three cyclo-dipeptides were also composed of QC intermediates. Fluorescent characterization demonstrated that the distinct morphologies presented different photoluminescence, with cyclo-HH nanofibers showing emission at 430 nm (Ex: 370 nm) (Fig. 7e), cyclo-FF nanoparticles at 530 nm (Ex: 450 nm) (Fig. 7f), and cyclo-YY nanorods at 570 nm (Ex: 480 nm) (Fig. 7g).

## Discussion
We demonstrate here that W-based aromatic cyclo-dipeptides self-assemble to dimeric QDs, which act as building blocks to further organize into supramolecular structures. Due to the extensive and directional QC regions within, the assemblies show intrinsic photoluminescence properties. Especially, through amino acid substitution, metal ion coordination, molecular oxidation or UV irradiation, and solvent replacement, the supramolecular morphologies could be finely controlled, ranging from dimeric QDs to larger organizations. Correspondingly, the photoluminescence could be tuned, covering most of the visible into the NIR spectral region. The biocompatibility and wide-spectrum emission features make these supramolecular structures highly suitable for in vivo bio-imaging applications with no detected cytotoxicity and for the fabrication of LEDs, where the assemblies are used as phosphors. Finally, further exploration is expected to demonstrate the potential of these self-assemblies for diverse applications. Thus, a variety of modulation strategies, such as the design of larger cyclo-oligopeptides (cyclo-tripeptides, cyclo-tetrapeptides, etc.), flexible assembly approaches (co-assembly, covalent conjugation), introduction of more complicated

reactions and self-assembly under different conditions (using physical vapor deposition), can be employed to tune the self-assembly and photoluminescence of cyclo-peptides.

Metal ions play important roles in peptide self-assembly processes. The cyclo-dipeptides QC self-assemblies described here might be used as fluorescent markers to probe the dynamics of aggregations processes, such as amyloidogenic polypeptide folding. This can supply a real-time, visual detection of the metabolic activities[58], and even facilitate screening for inhibitors of these processes, acting by interaction with metal ions. In addition, their nanoscale sizes, opto-electric properties, and intrinsic biocompatibility allow these nanostructures to be implanted into neuronal cells in order to investigate the interface between the structures and neurons[59]. Therefore, the metabolic procedures of the QC assemblies in neural cells and their response and influence on neuronal activities (such as current formation, synaptic activities, and signal transduction) can be studied by tracking their signals. This can provide the basis for diagnosis and treatment of sensory functions.

To conclude, based on comprehensive analyses of aromatic cyclo-dipeptides self-assemblies, as presented here, these materials could serve as bioinspired, organic, supramolecular alternatives to complement the state-of-the-art inorganic counterparts, and further broaden the application of QC materials.

## Methods
**Cells lines.** The MCF-7 human breast cancer cell line was purchased from the American Type Culture Collection (ATCC). The B16-BL6 murine melanoma cell line was obtained from the National Cancer Institute-Central Repository. The human skin HaCaT keratinocyte cell line, a transformed human epidermal cell line, was obtained from Dr. Norbert Fusenig of the Germany Cancer Research Center.

MCF-7 cells were cultured in Eagle's Minimum Essential Medium supplemented with 10% FBS. B16-BL6 and HaCaT cells were cultured in Dulbecco's Modified Eagle's Medium supplemented with 10% FBS. All culture media contained 1% penicillin and cells were maintained at 37 °C in a humidified 5% $CO_2$ incubator. Cells were periodically examined to verify the absence of mycoplasma contamination using the commercial detection kit (Lonza, Switzerland, LT07-703).

**Mice and care.** Athymic male NU/NU nude mice (6-week old) were purchased from Charles River (Wilmington, MA, USA) and maintained on a 16:8 h light–dark cycle. All procedures for animal use were approved by the Institutional Animal Care and Use Committee (IACUC) at The Ohio State University.

**Materials.** Cyclo-FW, cyclo-WW, cyclo-YY, and cyclo-FF were purchased from Bachem (Bubendorf, Switzerland), cyclo-Fw and cyclo-HH from DGpeptides (Hangzhou, China), anhydrous zinc chloride ($ZnCl_2$), copper chloride dihydrate ($CuCl_2 \cdot 2H_2O$), and anhydrous MeOH from Sigma Aldrich (Rehovot, Israel), DMSO from Sigma-Aldrich (St. Louis, MO). All materials were used as received without further purification. Water was processed using a Millipore purification system (Darmstadt, Germany) with a minimum resistivity of 18.2 MΩ cm.

**Sample preparation.** Cyclo-dipeptides were added to anhydrous MeOH, MeOH solutions of $ZnCl_2$ ($CuCl_2$), or DMSO solution of $ZnCl_2$, to final concentrations of 5.0 mM cyclo-dipeptides and 10.0 mM metal salts. To dissolve the peptides, the solutions were incubated in a water bath at 70 °C for 5 min, after which the solutions became transparent.

**Fluorescence.** UV-Visible fluorescence characterization: 600 μL sample solution was pipetted into a 1.0 cm path-length quartz cuvette, and the spectrum was collected using a FluoroMax-4 Spectrofluorometer (Horiba Jobin Yvon, Kyoto, Japan) at ambient temperature. For molecular fluorescence, the emission wavelength was set at 350 nm with a slit of 3 nm, and the excitation wavelength was set at 200–330 nm with a slit of 3 nm. For self-assembly fluorescence, the excitation wavelength was set at 370 nm with a slit of 3 nm, and the emission wavelength was set at 380–700 nm with a slit of 3 nm. For excitation-dependent maximal emission evolution experiment, the excitation wavelengths were set at 300–450 nm with a slit of 3 nm, and the emission wavelengths were set at 380–600 nm with a slit of 3 nm. According to the samples, anhydrous MeOH or MeOH solution of $ZnCl_2$ ($CuCl_2$) was used as background and subtracted. At least five measurements were performed and averaged for accuracy.

NIR fluorescence characterization: Cary Eclipse Fluorescence Spectrophotometer (Agilent Technologies) was used for measuring fluorescence excitation and emission spectra, with both slits width set at 5 nm. Three measurements were performed and averaged for accuracy.

**Fluorescent decay measurement (lifetime).** Six hundred microliter sample solution was pipetted into a 1.0 cm path-length quartz cuvette, and the spectrum was collected using a FluoroMax-4 Spectrofluorometer (Horiba Jobin Yvon, Kyoto, Japan) equipped with a NanoLED laser excitation source at ambient temperature. The wavelength was set as the maximum excitation and emission of the samples, and a LUDOX sample (silica beads, 2 μm) was used as the prompt. The lifetime was determined by fitting the fluorescent decay data from the DAS6 Analysis software (Horiba Jobin Yvon, Kyoto, Japan). Three measurements were performed and averaged for accuracy.

**Fluorescence photobleaching.** The kinetic measurements of Cy5.5 (excitation: 650 nm, emission: 720 nm), ICG (excitation: 785 nm, emission: 826 nm) and cyclo-WW + Zn(II) in DMSO (excitation: 550/680/770 nm, emission: 615/712/817 nm) were conducted using a Cary Eclipse Fluorescence Spectrophotometer (Agilent Technologies). The fluorescence emission data was collected at 80 points per second for 10 min. The excitation and emission slit widths were each set at 5 nm. At least three measurements were performed and averaged for accuracy.

**Fluorescence photostability.** Fluorescence emission spectra were measured at different time points (1 day, 7 days, and 14 days). The excitation and emission slit widths were each set at 5 nm. Three measurements were performed and averaged for accuracy.

**UV-Vis absorption.** For cyclo-dipeptides solutions absorption, 45 μL of the sample solution was pipetted into a 96-well UV-Star UV transparent plate (Greiner BioOne, Frickenhausen, Germany), and the UV-Vis absorbance was recorded using a Biotek Synergy HT plate reader (Biotek, Winooski, VT, USA), with a normal reading speed and calibration before reading. Anhydrous MeOH was used as background and subtracted. Three measurements were performed and averaged for accuracy.

**QD radius calculation.** The QD radius was calculated based on the model of organic QDs[34], as shown in Eq. 1:

$$R = \pi r_B^0 \sqrt{\frac{m_0/M}{\frac{\mu}{m_0 \varepsilon_\infty^2} - \frac{E_{ex}^{QD}}{R_y}}} \qquad (1)$$

Where $r_B^0 = \hbar^2 / m_0 e^2 = 0.53$ Å is the Bohr radius of the hydrogen atom; $m_0$ is the free electron mass ($9.11 \times 10^{-31}$ kg); $R_y = m_0 e^4 / 2\hbar^2 = 13.56$ eV is the Rydberg constant; $M = m_e + m_h$ is the translation mass of the exciton ($m_e$ and $m_h$ are the effective mass of electron and hole, respectively); $\mu = m_e m_h (m_e + m_h)$ is the reduced exciton mass; $\varepsilon_\infty$ is the high-frequency dielectric constant of the QD; $E_{ex}^{QD}$ is the exciton binding energy.

Since obtaining the accurate reflective index of peptide QDs is technically challenging, the refractive index of the analogous benzene crystal, namely $n = 1.5$, was used to calculate $\varepsilon_\infty$. This defines $\varepsilon_\infty = n^2 = 2.25$.

The optical absorption starts from $\lambda_{ion}$ 250 nm ($\hbar\omega_{ion} = 4.96$ eV) (Fig. 1c), corresponding to the breaking of the binding exciton state. The value of $\hbar\omega_{ion}$ corresponds to the QD energy gap. The difference between $\hbar\omega_{ion}$ and the phononless line $\hbar\omega_g^0 = 4.59$ eV equals 0.37 eV, representing the exciton binding energy, $E_{ex}^{QD}$ of the QD.

The effective masses of electrons and holes are almost identical and close to $0.5m_0$. Consequently, for $\mu = 0.5m_e = 0.25m_0$ and $M = M_0$, the QD diameter of the cyclo-dipeptides was calculated to be D ≈ 2.24 nm, approximately the dimension of a dimer.

For Job Plot analysis of cyclo-WW + Zn(II), a fixed total concentration of 15.0 mM was used, with the following molar proportions (corresponding concentrations) of cyclo-WW: 0.0 (0.0 mM), 0.1 (1.5 mM), 0.2 (3.0 mM), 0.3 (4.5 mM), 0.4 (6.0 mM), 0.5 (7.5 mM), 0.6 (9.0 mM), 0.7 (10.5 mM), 0.8 (12.0 mM), 0.9 (13.5 mM), 1.0 (15.0 mM). One milliliter sample of each solution was pipetted into a 1.0 cm path-length quartz cuvette, and a T60 visible spectrophotometer (PG Instruments, Leicestershire, United Kingdom) was used for spectra collection with a fixed spectral bandwidth of 2 nm and a 200–800 nm wavelength range. Anhydrous MeOH solutions of $ZnCl_2$ at the corresponding concentrations were used as background and subtracted. The absorbance at 515 nm was extracted to generate the Job Plot vs. molar proportions of cyclo-WW.

**Theoretical calculations of molecular orbital amplitudes and energy levels.** Density functional theory calculations were carried out based on the self-consistent solution of Kohn–Sham function and the projector augmented wave pseudopotential as implemented in Vienna Ab-initio Simulation Package (VASP). The exchange-correlation potential is in the form of Perdew-Burke-Ernerhof (PBE) with generalized gradient approximation (GGA). For the structural relaxation, the energy convergence threshold was set to $10^{-5}$ eV and the residual force on each atom was less than 0.03 eV Å$^{-1}$. The cutoff energy for the plane-wave basis was set to 500 eV. To eliminate interaction between the molecule and its periodic images, a vacuum distance larger than 15 Å for each direction in the supercell geometry was used.

**Nucleic magnetic resonance (NMR).** Cyclo-WW or cyclo-WW + Zn(II) were dissolved in deuterated solvent with tetramethylsilane as the internal standard to prepare sample solutions with 5.0 mM dipeptide and 10.0 mM Zn(II). $^1$H NMR spectra were recorded on a Bruker AV-400 NMR spectrometer with chemical shifts reported as p.p.m. The difference in the chemical shifts values before and after the addition of Zn(II) into the cyclo-WW solution ($\Delta\delta = (\delta_{(cyclo-WW)} - \delta_{(cyclo-WW+Zn(II))})$) were calculated in ppb and plotted as a function of amide, aromatic, and aliphatic protons.

**Mass spectrometry (MS).** The MS experiment was performed using a LCMS Xevo-TQD system including an Acquity model UPLC and a triple quad mass spectrometer (Waters, Massachusetts, USA). The positive electrospray ionization (ES+) channel was used for analysis.

**Scanning electron microscopy (SEM).** Twenty microliter solution samples were placed onto a clean glass slide and allowed to adsorb for a few seconds. After removing excessive liquid with filter paper, the slide was coated with Cr and observed under a JSM-6700 field emission scanning electron microscope (JEOL, Tokyo, Japan) operated at 10 kV.

**Quantum yield (QY) measurement.** Five milliliter sample solution was pipetted into a 1.0 cm path-length quartz cuvette with a tube (Hellma, Müllheim, Germany), and the absolute quantum yield was measured on an absolute PL quantum yield spectrometer C11347 (Hamamatsu Photonics, Shizuoka, Japan) at ambient temperature. The relevant blanks, namely anhydrous MeOH, MeOH solutions of $ZnCl_2$ ($CuCl_2$), or DMSO solution of $ZnCl_2$, were used as background and subtracted. At least five measurements were performed and averaged for accuracy.

**Dynamic light scattering (DLS)**. Eight hundred and fifty microliter of the sample solution was introduced into a DTS1070 folded capillary cell (Malvern, Worcestershire, U.K.), and the size was measured using a Zetasizer Nano ZS analyzer (Malvern Instruments, Malvern, UK) at 25.0 °C and a backscatter detector (173°). Three measurements were performed and averaged for accuracy.

**Fourier transform infrared spectroscopy (FTIR)**. Seven hundred and fifty microliter of the sample solution was dropped onto polyethylene IR card (International Crystal Labs, Garfield, New Jersey, USA) and air drying. The FTIR spectra were recorded on a Nicolet 6700 FTIR spectrometer (Thermo Scientific, Waltham, Massachusetts, USA), from 4000 to 400 $cm^{-1}$ at room temperature. One hundred and twenty-eight scans were collected with a spectral resolution of 4 $cm^{-1}$ in nitrogen atmosphere. Corresponding reference spectra (anhydrous MeOH or MeOH solution of $ZnCl_2$ ($CuCl_2$)) were recorded under identical conditions and subtracted.

**Atomic force microscopy (AFM)**. The cyclo-dipeptide self-assemblies were characterized using an MFP-3D AFM system with an ACTA-50 Probe (AppNano). Pyramidal shaped silicon tip was used, with radius of curvature below 10 nm, alumina reflex coating, a spring constant of 13–77 N $m^{-1}$, and a frequency of 200–400 kHz. The Igor Pro software in AC mode was used for image recording, thereby minimizing the distortion caused by mechanical interactions between the surface and the AFM tip.

**Transmission electron microscopy (TEM)**. A carbon-coated copper grid was placed on a 10 μL cyclo-dipeptide droplet for 1 min and then blotted. Next, the washed grid was placed on a 10 μL droplet containing 4% (w/v) uranyl acetate solution for 1 min and then blotted. Samples were examined using Tecnai G2 Spirit TEM (FEI) at 80 kV.

**X-ray diffraction (XRD)**. Spectra were recorded using a Bruker D8 Advance X-ray powder diffractometer (Bruker) at room temperature with a scan range 2θ of 5–45° and a count of two seconds. Cyclo-WW + Zn(II) nanospheres powder was placed on the standard flat sample reflection holder. Data collection and analysis was performed using the MDI Jade software.

**Fabrication and characterization of LEDs**. Commercially available InGaN chips were used at the bottom of the LED base. For the preparation of the color conversionlayer, the cyclo-WW + Zn(II) MeOH solution was purged dry using ultrapure nitrogen, and then mixed with PDMS at a mass ratio of 1:1. The mixtures were applied on InGaN chips and after curing at 80 °C for 1 h, the LEDs peptides phosphors were obtained.

**Cytotoxicity**. HaCaT, B16-BL6, and MCF-7 cells were seeded in a 96-well plate with 100 μL of culture medium per well and incubated at 37 °C in 5% $CO_2$ for 12 h to allow the cells to adhere to the surface. The medium was then replaced with a medium containing cyclo-WW + Zn(II) nanospheres at two different concentrations (62.5, 125.0 nM in cell culture medium), or with naïve medium as a control. Cells were incubated for 24 h before determining cell viability using the CCK-8 assay (Dojindo Molecular Technologies), according to the manufacturer's instructions. The absorbance at 490 nm was determined using Opsys microplate reader (Dynex Technologies, Chantilly, VA).

**In vivo NIR imaging**. Cyclo-WW + Zn(II) nanospheres (50 μL, 2.7 mM diluted in water) were administered into nude mice by subcutaneous injection. Whole body NIR fluorescence imaging was conducted with the mice anesthetized (2.5% isoflurane in oxygen flow, 1.5 L $min^{-1}$) immediately following the injection using an IVIS Spectrum Imaging System (PerkinElmer). All images were taken using emission filters designed for DsRed (575-650 nm), Cy5.5 (695–770 nm), and ICG (810–875 nm), with excitation wavelengths of 535 nm, 640 nm, and 745 nm, respectively. The fluorescent light emitted from the mice was detected by a CCD camera. Data acquisition and analysis was performed using the Living Image 4.2.1 software.

**Data availability**. The authors declare that the data supporting the findings of this study are available within the article and its Supplementary Information file, or are available from the authors upon request.

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

## Acknowledgements

This work was supported in part by the European Research Council under the European Union Horizon 2020 research and innovation program (no. 694426) (E.G.), HUAWEI company (E.G.), the NIH research grant (R01-CA226251) (M.Z.), the Office of Naval Research (ONR: M.Z.), the Research Supported by the CAS/SAFEA International Partnership Program for Creative Research Teams (J.L.), and the Ohio State University Comprehensive Cancer Center (Pelotonia Postdoctoral Fellowship to Z.F.). The authors thank Dr. Miri Kazes for experimental assistance, Dr. Sigal Rencus–Lazar for language editing, and the members of the Zhang, Gazit and Li laboratories for helpful discussions.

## Author contributions

M.Z., E.G., and J.L. conceived and designed the work; K.T. and Z.F. conducted SEM, AFM, FTIR, and MS characterizations; K.T. and P.M. conducted measurements of visible fluorescence, UV-Vis absorbance and NMR; Z.F. conducted measurements of TEM and NIR fluorescence; Z.T. and S.N.Q. performed LED fabrication and characterizations; K.T. and R.A. conducted lifetime measurements; M.S.S. performed theoretical calculations; L.S., M.R., S.S.-N., D.H., Z.P., S.G., J.M., and F.Y. performed cytotoxicity test and in vivo bio-imaging experiments. K.T. and Z.F. coordinated all the work, analyzed the results, wrote, and edited the manuscript. All authors discussed and commented on the manuscript.

## Additional information

**Competing interests:** The authors declare no competing interests.

