## [Peer Review File · Nature Communications]

Reviewers' comments:

Reviewer #1 (Remarks to the Author):

This manuscript reports photoluminescent peptide assemblies and their applications for biological imaging. The manuscript is premature to be published as there is significant a lack of clarity in the manuscript.

First of all, there is no any clear explanation what is the nature of the luminescence of monomers, dimers and peptide assemblies. If these are quantum confined structures, the authors must provide appropriate theoretical models and necessary calculations proving that. Meanwhile, the authors write "Due to the lack of well-known electronic properties of peptide QDs", in the manuscript and at the same time they make some assumptions and approximation without knowing what kind of electronic parameters they deal with here.

Secondly, there is no any information on the fluorescent life times of the monomers, dimers, peptide assemblies and modulated structures. Without these data it is pointless to discuss mechanisms of the fluorescence modulation using coordination with Zn(II), substitution, oxidation by Cu(II) or UV irradiation, and replacement of the solvent or side-chain aromatic moieties. Currently, all discussions on the fluorescence modulation are purely speculative in the manuscript. Appropriate FLIM studies would also be very useful to have .

Thirdly, the authors only show the structures of the individual molecules but it is absolutely unclear what the actual structures of the aggregates are? How are the molecules packed and interact in them? This is extremely important to know as the aggregates do all "magic" in all this work.

Finally , there are some concerns about the biological studies. According to results of the cytotoxicity tests, most of the viability values with error bars exceed 100 % (Figure 5g). How can it happen? Something is wrong with the testing and it should be re-done.

Reviewer #2 (Remarks to the Author):

This work reports some observation based on self assembly of aromatic cyclo-dipeptides under zinc or copper ion or by UV radiation favoring emissions in the near infrared. Such derivatives were used in some in vivo bio-imaging. The bio-imaging part is carried out meticulously however; the materials used as probes are subject to scrutiny. The identification of the aggregated molecules questions their purity. Random synthesis of nano molecules will obviously show their photoluminescence properties but the inherent question could be as how pure the nano molecules are. Is the experimental procedure reproducible? The discrepancies present here are many and some of these are mentioned here. Number one 1: what is origin of the peak at 515 nm in Zn complex with cyclo WW dimer (Figure 4a) . This charge transfer origin of absorption should require a bit more clarification. This specially true as the complex formation is thought to be in the ratio as one dimer : one zinc (Job's method). It is essential to isolate this adduct in pure form and then subjected to study further. It is also argued that cyclo-dipeptides indeed self-assembled in methanol then the result shown in Figure S-7 contradicts this observation as here also the possibility of self aggregation exists or the zinc ion inhibit self aggregation to dimer formation? Number 2: The FT-IR is all the time taken from the mixture solution. Is it true that the said conversions are 100% ? One should try to understand that such IR absorptions will be additive when performed from mixture and assignments of vibration modes from such non stoichiometric mixtures would be wrong. Number3: The characterization by mass spectroscopy to be analyzed with caution as the energy used in such technique could be enough to derive further reactions. This specially true with metal ions. Fitting with metal ion , say cuprous with a fragment may not lead to the identity of the complex under study. For example copper is toxic and using copper as oxidant to facilitate oxidative aggregation would allow the reduced copper to stay back there. There has been no assurance in the recipe to get rid of the oxidized aggregate freed from any

copper ion.

Thus overall the purity of the probe used for such study is severely questionable. Until and unless these are isolated in pure form and characterized properly, the reported result may be faulty and not reproducible. Therefore the paper should not be published.

Dr. Sabyasachi Sarkar

Professor Emeritus (Honorary)

IIESTS, Nanoscience and Synthetic Leaf Laboratory at Downing Hall

Botanic Garden -711103, West Bengal, India

Reviewer #3 (Remarks to the Author):

In this work is presented the formation of quantum confined materials by self-assembly of cyclo dipeptides building blocks. The fluorescence properties are modulated by self-coordination with metals.

The obtained nanomaterials exhibit interesting properties as stability against photobleaching and also exhibit near infrared fluorescence which is relevant for in vivo applications.

The characterization of the nanomaterials is well done, however the biological assays are not well described and discussed. Statistics of in vivo experiments is not well detailed. The experiments in vitro and in vivo are very preliminar to consider this nanomaterials for a potential biomedical application.

There are important issues to be solved before publication:

-How can be modulated the size of the nanoparticles?. It is important to discuss how can be controlled the size of the nanoparticles obtained in the different conditions.

- What happen whether the concentrations of the cyclopeptides or the concentration of the metals are modified in figure S17 or Figure 6?

-Which is the zeta potential of the nanoparticles?. This point is relevant for future applications.

-In the in vivo and in vitro experiments which is the concentration of nanoparticles that have been tested?

-The nanoparticles in which solvent have been administrated?

-Is it possible to functionalize the obtained nanoparticles for targeting? Please discuss this point.

-This nanoparticles can be useful for photodynamic therapy?. By irradiation is possible to form radicals in the in vivo conditions?

-Which is the stability of the nanoparticles in cell culture media or in vivo?

-Are the metal or the molecules released from the nanoparticles in the biological media?

On the other hand, in the conclusion section it is mentioned an application for neuronal cells, please explain this point. Is not clear.

A point-by-point response to the reviewers' comments

We would like to thank the reviewers for their evaluation, comments and suggestions. The following is a list of changes made in the revised manuscript according to the comments made by the reviewers. We believe that the revised manuscript indeed provides a better and clearer account of the reported work. For clarity, our responses are denoted in red.

Reviewer #1 (Remarks to the Author):

This manuscript reports photoluminescent peptide assemblies and their applications for biological imaging. The manuscript is premature to be published as there is significant a lack of clarity in the manuscript. First of all, there is no any clear explanation what is the nature of the luminescence of monomers, dimers and peptide assemblies. If these are quantum confined structures, the authors must provide appropriate theoretical models and necessary calculations proving that. Meanwhile, the authors write "Due to the lack of well-known electronic properties of peptide QDs", in the manuscript and at the same time they make some assumptions and approximation without knowing what kind of electronic parameters they deal with here.

Response: We thank the reviewer for the careful reading and for raising these issues, allowing us to add important data and further clarify our analysis. As the reviewer stated, theoretical calculations can be extremely valuable for clarifying the molecular mechanisms underlying the dipeptides self-assembly and the corresponding photoluminescent properties. For this purpose, we added density function theory (DFT) calculations to the revised manuscript. This analysis demonstrated that the spatial distributions of the highest occupied molecular orbitals and the lowest unoccupied molecular orbitals of the cyclo-dipeptides are mainly concentrated in the tryptophan (W) side-chain indole rings (Fig. 1f). Specifically, the band gaps (ΔE) of cyclo-FW and cyclo-WW were calculated to be 3.63 eV and 3.56 eV, respectively, indicating their wide-gap semiconductive nature (*J. Chem. Phys.* **134**, 175101 (2011); *J. Chem. Phys.* **140**, 124511 (2014)). This suggests that the dimerization was mostly driven by the π - π interactions between aromatic side-chains, especially the indole rings. In addition, the NMR results demonstrated the upfield shifting of the hydrogen atoms at the indole rings after coordination with Zn(II) (Fig. 4c, Fig. S11), indicating the elevated delocalization of the π electrons on the aromatic rings. Combined with the theoretical calculations, these results demonstrate that the indole rings formed aromatic interactions with each other and through-space conjugation of electrons took place (*J. Am. Chem. Soc.* **139**, 16264-16272 (2017); *J. Am. Chem. Soc.* **139**, 17882-17889 (2017)), underlying the molecular basis for quantum confined regions (*J. Am. Chem. Soc.* **132**, 15632-15636 (2010)).

For QD radius calculation using UV-vis absorption, several physical parameters, such as the refractive index and high-frequency dielectric constant, are required (as detailed in the "Materials and Experimental Section" in the main text). Due to the intrinsic difficulty to purify the QDs and to experimentally measure their physical parameters, the refractive index of the other organic materials similar to short aromatic peptides, such as benzene crystals with $n = 1.5$, was used for calculation.

Following the reviewer's comments, we added the theoretical calculation, NMR results and relevant references, and revised the manuscript as follows:

Page 5, Line 13: Therefore, it can be concluded that the cyclo-dipeptide monomers first formed dimers, which behaved as QDs and served as the fundamental building blocks to further organize into supramolecular structures (Fig. 1e). Theoretical calculations using density function theory demonstrated that the spatial distributions of the highest occupied and lowest unoccupied molecular orbitals of the dipeptides were mainly concentrated on the side-chain indole rings (Fig. 1f). Specifically, the band gaps (ΔE) of cyclo-FW and cyclo-WW were calculated to be 3.63 eV and 3.56 eV, respectively, indicating their wide-gap semiconductive nature^{36, 37}. This implies that the dimerization was mostly driven by π - π interactions between aromatic side-chains, especially the

indole rings. Particularly, the aromatic interactions could induce the through-space conjugation of the electron clouds from adjacent indole rings, thus restricting the molecular motions^{38, 39} and underlying the molecular basis for QC regions³⁴. The QC effects and restricted molecular activities resulted in the release of excitation energy exclusively as emitted light^{10, 12}.

Page 10, Line 3: The ¹H-NMR spectra revealed that after coordination, the hydrogen atoms in the backbone showed downfield shifting (Fig. 4c, Fig. S11), indicating that the shielding effect became weaker. This suggests that the diketopiperazine ring contributed to the coordination by supplying electrons to interact with Zn(II)⁵². In contrast, the hydrogen atoms in the indole rings showed upfield shifting (Fig. 4c, Fig. S11), indicating that the delocalization of the π -electrons on the aromatic rings became stronger⁵². This demonstrates that the indole rings formed aromatic interactions with each other and through-space conjugation of electrons took place, consistent with the theoretical calculations.

Page 22, Line 3: Due to the technical difficulty to obtain the accurate reflective index of peptide QDs, we used the refractive index of the similar benzene crystal, with $n = 1.5$, for calculating ϵ_{∞} . This defines $\epsilon_{\infty} = n^2 = 2.25$.

Page 6, Figure 1: **Fig. 1 Cyclo-dipeptides dimeric QDs serve as building blocks for self-assembly into QC supramolecular structures.** **a**, Molecular structures of cyclo-FW and cyclo-WW. **b**, Excitation spectra of cyclo-dipeptide monomers (thin light curves, 0.05 mM) and self-assemblies (thick dark curves, 5.0 mM) in MeOH. The excitation wavelengths red shifted from 285 nm to ~305 nm after self-assembly. **c**, UV-Vis absorption spectra of the cyclo-dipeptides after self-assembly, showing spike-like absorptions at 273 nm, 280 nm and 289 nm, characteristic of the formation of QD structures. **d**, MS spectra of cyclo-FW and cyclo-WW at 5.0 mM in MeOH. The dimeric MW is marked in red, and the monomeric MW is marked in black. **e**, Schematic representing the process of cyclo-dipeptides self-assembly: the monomers form dimeric QDs, which serve as the building blocks

to self-assemble into larger QC architectures. **f**, Calculated molecular orbital amplitude plots and energy levels of the highest occupied and lowest unoccupied molecular orbitals of cyclo-FW and cyclo-WW, showing band gaps of 3.63 eV and 3.56 eV, respectively.

Secondly, there is no any information on the fluorescent life times of the monomers, dimers, peptide assemblies and modulated structures. Without these data it is pointless to discuss mechanisms of the fluorescence modulation using coordination with Zn(II), substitution, oxidation by Cu(II) or UV irradiation, and replacement of the solvent or side-chain aromatic moieties. Currently, all discussions on the fluorescence modulation are purely speculative in the manuscript. Appropriate FLIM studies would also be very useful to have.

Response: We thank the reviewer for raising this issue, allowing us to add important experimental data to the revised manuscript. As the reviewer mentioned, fluorescent lifetime reflects the stability of the excited electrons. Therefore, the fluorescent decay times of the peptides self-assemblies should change upon coordination with Zn(II), amino acid substitution, oxidation by Cu(II) or UV irradiation or replacement of the solvent. The comparison between the lifetimes of the different systems can help to elucidate the mechanisms underlying fluorescence modulations. Specifically, the comparison between the lifetimes of cyclo-WW in MeOH and cyclo-WW+Cu(II) / cyclo-WW+UV can reflect the redox reactions, while the comparison between the lifetimes of cyclo-WW in MeOH and cyclo-WW+Zn(II) can indicate the influence of coordination-induced electron transfers on fluorescent decay. In addition, the comparison between the lifetimes of cyclo-WW in MeOH and in DMSO can demonstrate the effects of aggregation on the stability of the excited electrons. These analyses are therefore highly instrumental for modulating the fluorescence of dipeptides self-assemblies and exploring their potential applications.

Following the reviewer's comment, we added the fluorescence lifetime results and revised the manuscript as follows:

Page 8, Line 6: HPLC analysis confirmed that the conversion was complete and the oxidized cyclo-WW was pure (Fig. S7). Fluorescent decay experiments demonstrated that after oxidation, the lifetime increased (6.3 ns for cyclo-WW+Cu(II), 8.0 ns for cyclo-WW+UV) compared to cyclo-WW (5.6 ns) (Fig. 2f, Fig. S8a), confirming that the redox reactions indeed took place.

Page 9, Line 17: In contrast, no new band appeared when Zn(II) was introduced into a monomeric cyclo-WW solution (Fig. S10), confirming that the dimers were indeed the form that complexed with Zn(II). Especially, the charge transfer could deliver the excited electrons, resulting in reduced fluorescence decay time, from 5.6 ns of cyclo-WW to 3.6 ns (Fig. 2f, Fig. S8a).

Page 14, Line 13: Specifically, the peaks at 615 nm, 712 nm and 817 nm were emitted using excitation wavelengths between 520 and 780 nm (Fig. 5c; Fig. S23), with lifetimes of 11.1 ns, 10.0 ns and 5.1 ns, respectively (Fig. 2f, Fig. S8b), suggesting that as aggregation progressed, the excited electrons became more stable. This indicates a promising potential of the supramolecular structures to be used as bio-inspired alternatives for stable, long-term imaging.

Page 8, Figure 2: **Fig. 2 Visible fluorescence properties and modulation of cyclo-dipeptides self-assemblies.** **a-e**, Fluorescent emission of: **a**, cyclo-FW and cyclo-WW self-assemblies. **b**, cyclo-WW+Zn(II). **c**, cyclo-Fw. **d**, cyclo-WW+Cu(II). **e**, cyclo-WW+UV. The insets show the corresponding solutions under UV light (365 nm). In (**b-e**), intrinsic fluorescent emissions of the corresponding cyclo-dipeptides were added and marked in black for comparison. **f**, Lifetime statistics of cyclo-WW self-assemblies. Left panel (red): cyclo-WW in MeOH (cyclo-WW: 370/425; + Zn(II): 370/520; + Cu(II): 370/465; + UV: 370/465). Right panel (blue): cyclo-WW+Zn(II) in DMSO (540/610; 670/712; 740/817).

Thirdly, the authors only show the structures of the individual molecules but it is absolutely unclear what the actual structures of the aggregates are? How are the molecules packed and interact in them? This is extremely important to know as the aggregates do all “magic” in all this work.

Response: We thank the reviewer for raising this issue. Following our response to the first question of the reviewer, in fact, several recent publications reported that through-space conjugation of aromatic electrons during aggregation can decrease bandgaps and restrict the molecular motions (*J. Am. Chem. Soc.* **139**, 16264–16272 (2017); *J. Am. Chem. Soc.* **139**, 17882–17889 (2017); *Adv. Mater.* **26**, 5429–5479 (2014)). This can result in the release of excitation energy through emission. Therefore, driven by the aromatic interactions, the dipeptides monomers dimerize into QDs, in which the aromatic integration domains form quantum confinements and the molecules are restricted to move (rotate or vibrate). The QDs then self-associate to crystals (cyclo-FW) or spherical nanoparticles (cyclo-WW) (Fig. 1e). For spherical nanoparticles, the molecules are believed to form sandwiched bilayers, with the backbone diketopiperazine rings forming hydrogen bonding and the aromatic side-chains forming aromatic interactions, while for crystals, crystallography can be used to precisely analyse the molecular structures. In fact, unpublished results, beyond the scope of the current manuscript, include the analysis of these crystal structures, demonstrating them to be organic semiconductors, showing unique optoelectric, mechanical, ferroelectric and piezoelectric properties.

Since the peptides self-assembly is greatly dependent on the experimental conditions, such as solvent polarity, metal ion coordination, redox, and amino acid residue substitutions, the self-assembled

morphologies and corresponding photoluminescence of W-containing cyclo-dipeptides can be modulated through changing the solvents (MeOH to DMSO), Zn(II) coordination, Cu(II) or UV oxidation and replacing indole rings with other aromatic side-chains (W to H, F, Y).

Following the reviewer's comment, we revised the manuscript correspondingly. Please see our response to the referee's first question in this letter.

Finally, there are some concerns about the biological studies. According to results of the cytotoxicity tests, most of the viability values with error bars exceed 100% (Figure 5g). How can it happen? Something is wrong with the testing and it should be re-done.

Response: We thank the reviewer for bringing this issue to our attention. We re-performed the cytotoxicity experiment accordingly, and the new results are presented in the revised manuscript.

The manuscript was modified to present the results of the repeated experiments, as follows:

Page 15, Figure 6c:

Reviewer #2 (Remarks to the Author):

This work reports some observation based on self-assembly of aromatic cyclo-dipeptides under zinc or copper ion or by UV radiation favoring emissions in the near infrared. Such derivatives were used in some in vivo bio-imaging. The bio-imaging part is carried out meticulously however; the materials used as probes are subject to scrutiny. The identification of the aggregated molecules questions their purity. Random synthesis of nano molecules will obviously show their photoluminescence properties but the inherent question could be as how pure the nano molecules are. Is the experimental procedure reproducible?

Response: We thank the reviewer for this comment, allowing us to clarify this experimental point. Like conventional amphiphiles, peptide self-assembly is a highly-ordered aggregation process. Driven by several driving forces (non-covalent interactions), the peptide molecules regularly organize into supramolecular structures with specific morphologies. Especially, these non-covalent interactions (mainly aromatic interactions and hydrogen bonding) can decrease the bandgap, thus resulting in the red-shifted photoluminescence of the self-assemblies (*Science*, **2017**, 358, eaam9756; *J. Am. Chem. Soc.* **2017**, 139, 16264–16272; *J. Am. Chem. Soc.* **2016**, 138, 3046–3057; *Adv. Mater.* **2014**, 26, 5429–5479). However, it should be mentioned that not all the self-assemblies show photoluminescence in the visible region. Thus, only assemblies with extensive and directional hydrogen bonding and aromatic interactions can show aggregation-induced visible emission behaviour (*Science*, **2017**, 358, eaam9756).

By controlling the subtle balance of these interactions, the self-assembly morphologies can be finely tuned between nanospheres, nanofibers, nanotubes, nanoplates and so on. Therefore, by modifying the self-assembly conditions, such as solvents, pH, ionic strength, temperature, etc., peptide self-assemblies, as well as their photoluminescence, can be accurately modulated and repeated. In fact, during the work, we strictly controlled the experimental conditions and every test was performed at least three times, showing excellent reproducibility.

Following the reviewer's comment, we emphasized the reproducibility of our findings in the "Materials and Experimental Section" and revised the manuscript as follows:

Page 20, Line 22: According to the samples, anhydrous MeOH or MeOH solution of ZnCl₂ (CuCl₂) was used as background and subtracted. **At least five measurements were performed and averaged for accuracy.**

Page 21, Line 9: The excitation and emission slit widths were each set at 5 nm. **At least three measurements were performed and averaged for accuracy.**

Page 23, Line 24: The relevant blanks, namely anhydrous MeOH, MeOH solutions of ZnCl₂ (CuCl₂), or DMSO solution of ZnCl₂, were used as background and subtracted. **At least five measurements were performed and averaged for accuracy.**

The discrepancies present here are many and some of these are mentioned here. Number one 1: what is origin of the peak at 515 nm in Zn complex with cyclo WW dimer (Figure 4a) . This charge transfer origin of absorption should require a bit more clarification. This specially true as the complex formation is thought to be in the ratio as one dimer : one zinc (Job's method). It is essential to isolate this adduct in pure form and then subjected to study further. It is also argued that cyclo-dipeptides indeed self-assembled in methanol then the result shown in Figure S7 contradicts this observation as here also the possibility of self aggregation exists or the zinc ion inhibit self aggregation to dimer formation?

Response: We thank the reviewer for allowing us to clarify this issue. The coordination of metal ion and ligands can decrease the transition energy of electrons, thus inducing a new absorption peak at longer wavelengths (*Chem. Rev.* **1982**, 82, 385-426). In our system, after complexing with Zn(II), a new absorption peak of cyclo-WW appeared at 515 nm, due to the peptide (ligand)-to-Zn(II) transfer. Especially, the new absorption resulted in colour change to pink from the original white/light yellow of the peptide itself (Fig. S9).

As the reviewer mentioned, the Job plot analysis demonstrated the stoichiometric ratio of cyclo-WW and Zn(II) to be 2:1, indicating that one dimer complexed with one Zn(II). Particularly, the AFM (Fig. 3c, Fig. S3), DLS (Fig. S4) and MS (Fig. S15) analyses confirmed this conclusion. However, extracting the dimers in a pure form is technically challenging. Since the dimers are only 2-3 nm in diameter (Figs. S3, S4), almost all

the purification strategies, such as centrifugation, size-exclusion chromatography or electrophoresis, cannot isolate them from monomers.

In spite of the inherent difficulty to purify the dimers, the exact molecular mechanism underlying cyclo-WW coordination with Zn(II) can still be clarified using NMR analysis. Comparison of the $^1\text{H-NMR}$ spectra of cyclo-WW and cyclo-WW+Zn(II) showed that the hydrogen atoms in the backbone diketopiperazine rings displayed downfield shifting after coordination (Fig. 4c, Fig. S11), indicating that Zn(II) attracted electrons from the backbone. This resulted in the screening of electron clouds to hydrogen protons becoming weaker, thus inducing downfield moving of chemical shifts. On the other hand, the chemical shifts of the hydrogen atoms in the side-chain indole rings showed upfield shifting (Fig. 4c, Fig. S11), indicating that the delocalization of π -electrons on the aromatic side-chains became stronger. Combined with the theoretical calculations (Fig. 1f) and FTIR (Fig. 4d) results, it can be concluded that the backbone diketopiperazine rings contribute to the coordination with Zn(II), and the indole rings form aromatic interactions. Therefore, a plausible molecular mechanism of cyclo-WW coordination with Zn(II) is depicted: the Zn(II) ion is embedded in a dimer of cyclo-WW, coordinating with two diketopiperazine rings, while the side-chain indole rings are located outside and form π - π interactions with each other (Fig. 4e).

As a control, there is an easier way to verify that dimers, rather than monomeric molecules, coordinate with Zn(II). At a very low concentration (0.05 mM), cyclo-WW did not self-assemble, retaining its monomeric state (Fig. 1b). Under these conditions, no absorption peak appeared around 515 nm after adding Zn(II) (Fig. S10), indicating that the cyclo-WW monomer did not complex with Zn(II).

Based on our findings, it can be concluded that in the absence of Zn(II), cyclo-WW monomers dimerize into dimeric QDs, which serve as building blocks to further self-assemble into larger nanospheres, while in the presence of Zn(II), the dimeric QDs complex with Zn(II), resulting in stabilization of the dimeric state. We speculate that the electrostatic repulsive interactions and the configurational changes upon coordination hindered the further assembly of the complexed dimers.

Following the reviewer's comment, we added the new NMR results and the relevant references, and revised the manuscript as follows:

Page 9, Line 12: Understanding the mechanisms underlying the modulation of the fluorescent properties of the self-assemblies is critical for their future applications⁵⁰. A new absorption peak at 515 nm, corresponding to ligand (peptide)-to-metal charge transfer⁵¹, emerged upon mixing cyclo-WW and Zn(II) in solution (Fig. 4a), indicating the formation of coordinated architectures. **Notably**, the new absorption resulted in a color change from the original white/light yellow of cyclo-WW to pink (Fig. S9).

Page 10, Line 3: The $^1\text{H-NMR}$ spectra revealed that after coordination, the hydrogen atoms in the backbone showed downfield shifting (Fig. 4c, Fig. S11), indicating that the shielding effect became weaker. This suggests that the diketopiperazine ring contributed to the coordination by supplying electrons to interact with Zn(II)⁵². In contrast, the hydrogen atoms in the indole rings showed upfield shifting (Fig. 4c, Fig. S11), indicating that the delocalization of the π -electrons on the aromatic rings became stronger⁵². This demonstrates that the indole rings formed aromatic interactions with each other and through-space conjugation of electrons took place, consistent with the theoretical calculations. Fourier-transform infrared spectroscopy (FTIR) characterizations demonstrated that in the presence of Zn(II), the N-H stretching vibration in diketopiperazine rings red-shifted to 3071 cm^{-1} from the original 3200 cm^{-1} ,⁵³ due to the decrease of the bond energies resulting from the metal ion adsorption (Fig. 4d, peak 1)⁵⁴, indicating that the nitrogen atoms in the backbone diketopiperazine rings contributed to the complexation with Zn(II). Combined with the NMR analysis, a plausible molecular mechanism of cyclo-WW coordination with Zn(II) is depicted in Fig. 4e. The Zn(II) ion is embedded in a dimer of cyclo-WW, coordinating with two diketopiperazine rings, while the side-chain indole rings form π - π interactions. It is possible that the complexation induces hindrance against

further aggregation of the dimers, thus resulting in the dimers separated from each other and showing stable photoluminescence regardless of the excitation wavelength.

Page 11, Figure 4: Fig. 4 Mechanisms underlying the modulation of cyclo-dipeptides self-assemblies fluorescence. **a**, UV-Vis absorption spectra of cyclo-WW self-assemblies in the absence or presence of Zn(II) in MeOH. A new peak at 515 nm appeared when complexing cyclo-WW and Zn(II). **b**, Job Plot analysis of cyclo-WW with Zn(II) at different ratios, with the total molar concentration fixed at 15.0 mM. The red lines were added for guideline, showing the intersection point at cyclo-WW proportion of 0.7. **c**, Chemical shifts of cyclo-WW hydrogen atoms upon coordination with Zn(II), compared to the peptide alone. The hydrogen atoms in different chemical environments are marked with italicized letters and labeled in (e). **d**, FTIR spectra of cyclo-WW self-assemblies in the absence or presence of metal ion/UV irradiation. The peaks of the active bonds are numerically marked. Note that the IR spectra were vertically moved for clarity. **e**, Schematic presentation showing the possible molecular mechanism of cyclo-WW dimer coordination with Zn(II): the backbone diketopiperazine rings contribute to the complexation through nitrogen atoms, while the side-chain indole rings form aromatic interactions. **f**, MS spectra of cyclo-WW+Cu(II) and cyclo-WW+Cu(II)+UV, showing the MW of oxidized cyclo-WW (marked in red) and reduced Cu(I), confirming the redox reactions in the solutions.

Number 2: The FTIR is all the time taken from the mixture solution. Is it true that the said conversions are 100% ? One should try to understand that such IR absorptions will be additive when performed from mixture and assignments of vibration modes from such non stoichiometric mixtures would be wrong.

Response: We thank the reviewer for raising this important point. The time-resolved fluorescence experiments demonstrated that cyclo-WW+UV emission reached the maximum after one week, while cyclo-WW+Cu(II) emission reached the maximum after one month (Fig. S12, Fig. S13), suggesting that the conversion dynamics was not slow, especially for cyclo-WW+UV. Actually, the HPLC analysis indicated the

high purity of the oxidized cyclo-WW in the cyclo-WW+UV system, showing only one main peak in the profile (Fig. S7A). Also, the fluorescent decay experiments demonstrated that the lifetime increased (6.3 ns for cyclo-WW+Cu(II), 8.0 ns for cyclo-WW+UV) compared to cyclo-WW itself (5.6 ns) (Fig. 2f, Fig. S8a), indicating that the redox reactions indeed took place.

In fact, the presence of metal salts (ZnCl_2 , CuCl_2) did not affect the FTIR spectra of peptides. The vibration peaks of metal-chloride bonds generally locate below 1000 cm^{-1} , while the vibration peaks of peptides bonds are above 1000 cm^{-1} (amide I region: 1700 cm^{-1} to 1600 cm^{-1} ; amide II region: 1600 cm^{-1} to 1500 cm^{-1}). Moreover, during the FTIR experiments, the corresponding backgrounds (ZnCl_2 or CuCl_2 solution at the same concentration) were always subtracted. Therefore, we strongly believe that the FTIR data are reliable.

Following the reviewer's comment, we added the HPLC and fluorescent decay data, and revised the manuscript as follows:

Page 8, Line 4: A similar emission was also observed after irradiation with UV light (365 nm) (cyclo-WW+UV) due to the UV-induced radical oxidation (Fig. 2e), implying a potential for photo-stimulated applications. HPLC analysis confirmed that the conversion was complete and the oxidized cyclo-WW was pure (Fig. S7).

Page 24, Line 5: 128 scans were collected with a spectral resolution of 4 cm^{-1} in nitrogen atmosphere. Corresponding reference spectra (anhydrous MeOH or MeOH solution of ZnCl_2 (CuCl_2)) were recorded under identical conditions and subtracted.

Number3: The characterization by mass spectroscopy to be analyzed with caution as the energy used in such technique could be enough to derive further reactions. This specially true with metal ions. Fitting with metal ion, say cuprous with a fragment may not lead to the identity of the complex under study. For example copper is toxic and using copper as oxidant to facilitate oxidative aggregation would allow the reduced copper to stay back there. There has been no assurance in the recipe to get rid of the oxidized aggregate freed from any copper ion.

Response: We thank the reviewer for allowing us to clarify this experimental point. As the reviewer mentioned, MS generally uses high energy which can possibly result in side reactions. Therefore, MS analysis alone is insufficient to confirm our conclusions. For this reason, in addition to MS, we performed further analyses to support the conclusion that cyclo-WW was indeed oxidized by Cu(II). For example, 1) When replacing Cu(II) with other oxidative metal ions, (Ag(I) or $[\text{AuCl}_4](-)$), the sample solutions showed the same emission spectra, with a maximum at 465 nm and blue-green colour under UV light (Fig. S16a). Especially, the intense redox reduced the metal ions to elementary metals (Fig. S16b); 2) When replacing cyclo-WW with cyclo-FW, the dipeptide did not complex with Zn(II) but showed enhanced fluorescence at 465 nm with a QY of 12% in the presence of Cu(II) (Fig. S17); 3) The fluorescent lifetime and FTIR experiments showed a distinct difference in the absence or presence of Cu(II) (Fig. 2f). These results confirmed that cyclo-WW was indeed oxidized by Cu(II), consistent with the MS analysis.

As the reviewer pointed out, the toxicity of copper/cuprous ions can limit the application of peptides self-assemblies, especially in biological systems. In fact, this was one of the reasons why we used cyclo-WW+Zn(II), rather than cyclo-WW+Cu(II), to study NIR photoluminescence and to perform *in vivo* bio-imaging. However, quantum confined peptide assemblies can be used in diverse fields, where cytotoxicity is not a pitfall, such as organic semiconductor devices (OLEDs, as demonstrated in the revised manuscript, or organic field-effect transistors).

Thus overall the purity of the probe used for such study is severely questionable. Until and unless these are

isolated in pure form and characterized properly, the reported result may be faulty and not reproducible. Therefore the paper should not be published.

Response: We thank the reviewer for further emphasizing the crucial point of the assemblies' purity, thus allowing us to expand our analysis and to address this central issue. Following the reviewer's comments, we supplemented diverse analyses, including theoretical calculations, HPLC, fluorescent decay, NMR and OLED applications of the quantum confined peptide assemblies. The molecular mechanisms underlying the self-assembly, along with photoluminescence, coordination with Zn(II) and the purity of oxidized cyclo-WW, were all addressed and clarified. Furthermore, we have validated our work to be reproducible and reliable. We believe that following the reviewers' valuable suggestions and comments, the revised manuscript indeed provides a better and clearer account of the reported work.

Reviewer #3 (Remarks to the Author):

In this work is presented the formation of quantum confined materials by self-assembly of cyclo dipeptides building blocks. The fluorescence properties are modulated by self-coordination with metals. The obtained nanomaterials exhibit interesting properties as stability against photobleaching and also exhibit near infrared fluorescence which is relevant for *in vivo* applications.

The characterization of the nanomaterials is well done, however the biological assays are not well described and discussed. Statistics of *in vivo* experiments is not well detailed. The experiments *in vitro* and *in vivo* are very preliminary to consider this nanomaterials for a potential biomedical application.

Response: We thank the referee for highlighting the self-assembling peptide quantum confined materials and for raising this issue. The purpose of our bio-imaging experiment was to demonstrate the suitability of the self-assembling dipeptide dots for *in vitro* and *in vivo* applications. The results showed that the fluorescent signals could keep stable and did not attenuate for one week, thus highlighting the possibility of utilizing the photoluminescent QC assemblies for *in vivo* bio-imaging. Therefore, our results lay the basis for potential future applications, such as tumour labelling, drug delivery/release tracking and even phototherapy. Our findings open the possibility of using self-assembling peptides for bio-imaging applications.

To further extend the application of the self-assembling peptides, we used the cyclo-WW+Zn(II) dots as phosphors to fabricate bio-inspired organic light-emitting diodes (OLEDs). The results showed that bright monochromatic green light could be illuminated (Fig. 6a, b), suggesting that the peptide dots can also be used for bio-inspired semiconductor-based devices. To the best of our knowledge, this is the first report of using peptides self-assemblies to construct OLED devices.

Following the reviewer's comment, we added the OLED results and revised the manuscript as follows:

Page 14, Line 28: The photoluminescent nature endows the peptide QC self-assemblies the ability to be used for photo-stimulated devices, such as OLEDs. By applying the mixtures of dried cyclo-WW+Zn(II) dots in MeOH and polydimethylsiloxane (PDMS) onto an indium gallium nitride (InGaN) chip, a prototypical OLED device using peptides self-assemblies as phosphors was fabricated (Fig. 6a). When applying voltages, bright green light was illuminated, as shown in the inset of Fig. 6a. Spectroscopic investigations demonstrated an emission around 550 nm regardless of the excitation wavelength (Fig. 6b), thus showing remarkable emission specificity. The red-shift of 30 nm from 520 nm (Fig. 2b) was probably due to aggregation that occurred during MeOH evaporation. To the best of our knowledge, this is the first report to use peptide self-assemblies to design OLEDs.

Page 15, Figure 6: **Fig. 6 Application of the cyclo-WW+Zn(II) QC self-assemblies.** **a**, Schematic presentation of the OLED setup using dried cyclo-WW+Zn(II) performed in MeOH as phosphors. The upper inset shows the working depiction of a prototype, emitting bright green light (Ex: 450 nm). **b**, Spectroscopic characterization of the OLED photoluminescence using three excitation wavelengths, showing the same emission at 550 nm. **c**, Cytotoxicity test of the cyclo-WW+Zn(II) nanospheres in DMSO (62.5 nM and 125 nM) towards B16-BL6, HaCaT and MCF7 cells. Viability relative to untreated controls \pm sd is shown. **d**, *In vivo* whole body NIR fluorescent imaging following subcutaneous injection of the nanospheres (50 μ L, 2.7 mM) into nude mice, showing notable emissions under various excitations. The dotted circle indicates the location of injection.

There are important issues to be solved before publication:

-How can be modulated the size of the nanoparticles?. It is important to discuss how can be controlled the size of the nanoparticles obtained in the different conditions.

Response: We thank the reviewer for allowing us to discuss this issue. Indeed, the photoluminescence wavelength of quantum confined materials is highly dependent on the particles size. Therefore, it is important and fundamental to control the peptide nanoparticle sizes before exploring their applications. Like conventional amphiphiles, peptide self-assembly is a highly-ordered aggregation process. Driven by several driving forces (mainly non-covalent interactions), the peptide molecules regularly organize into supramolecular structures with specific morphologies. By controlling the subtle balance between these interactions, the self-assembly morphologies can be finely tuned between nanospheres, nanofibers, nanotubes, nanoplates, and so on. Therefore, by modifying the self-assembly conditions, such as solvents, pH, ionic strength, temperature, etc., peptide self-assemblies can be accurately modulated.

Several methods can be used to modulate the size of the nanoparticles. 1) By changing the amino acid residues, such as replacement with other aromatic amino acids (Fig. 7) or *D*-type enantiomers (Fig. 3d), the self-assembling morphologies can be modified from nanospheres to nanofibers, nanoplates or nano-flower crystals; 2) By changing the solvents, such as MeOH to the more polar DMSO, the cyclo-WW + Zn(II) self-assembling nanoparticles can become larger (Fig. 3c vs Fig. 5a). Furthermore, when using water, a much

more polar solvent, the cyclo-dipeptides crystallize into needle-like crystals; 3) Through redox reactions, such as introduction of Cu(II) or UV irradiation, W-containing peptides can be oxidized, thus altering the self-assembling morphologies (Fig. 3e, f); 4) Through metal coordination, such as introduction of Zn(II) to complex with cyclo-WW, the self-assembly of cyclo-WW can halt at the dimer stage rather than forming larger nanospheres (Fig. 3c).

In addition, other modulation strategies, such as the design of larger cyclo-oligopeptides (cyclo-tripeptides, cyclo-tetrapeptides....), complexation with other metal ions, flexible assembly strategy (co-assembly, covalent conjugation of functional moieties), introduction of more complicated reactions and self-assembly under different conditions (solvents, physical vapour deposition), can be employed to tune the cyclo-dipeptides self-assemblies.

Following the reviewer's comment, we clarified this point and revised the manuscript as follows:

Page 17, Line 15: The biocompatibility and wide-spectrum emission features make these supramolecular structures highly suitable for *in vivo* bio-imaging applications with no detected cytotoxicity and for the fabrication of OLEDs, where the assemblies are used as phosphors. Finally, further exploration is expected to demonstrate the potential of these self-assemblies for diverse applications. Thus, a variety of modulation strategies, such as the design of larger cyclo-oligopeptides (cyclo-tripeptides, cyclo-tetrapeptides, etc.), flexible assembly approaches (co-assembly, covalent conjugation), introduction of more complicated reactions and self-assembly under different conditions (using physical vapor deposition), can be employed to tune the self-assembly and photoluminescence of cyclo-peptides.

- What happen whether the concentrations of the cyclopeptides or the concentration of the metals are modified in figure S17 or Figure 6?

Response: We thank the reviewer for raising this question. Our studies showed that aromatic cyclo-dipeptides first form dimers, which behave as quantum dots to further self-assemble into supramolecular structures (Fig. 1e). Therefore, the self-assembling morphologies and corresponding photoluminescence will alter when material concentrations are modified. At very low concentrations, where the cyclo-dipeptides are still monomers, there are no self-assembling nanostructures and the intrinsic fluorescence of the monomers is in the UV region (less than 400 nm, Fig. 1b). As concentrations increase, the monomers start to dimerize, and small nanoparticles, less than 5 nm in diameter, can be detected in AFM images (Fig. 3c, Fig. S24). Correspondingly, the aggregation-induced emission in the visible region appears. As the concentrations further increase, the dimeric QDs further aggregate into larger architectures (Fig. 5a, Fig. 7), and their photoluminescence red-shifts to a longer-wavelength region, even within the near infrared domain (Fig. 5c). When the concentrations are high enough, the peptides crystallize into needle-like crystals and precipitate.

The Job plot analysis demonstrated that the stoichiometric ratio between the cyclo-WW monomer and Zn(II) is 2:1, showing that one cyclo-WW dimer interacts with one Zn(II) ion (Fig. 4b). Therefore, if the concentration of Zn(II) is lower, cyclo-WW dimers will further self-assemble into larger nanospheres and show fluorescence at 425 nm, rather than the emission at 520 nm due to Zn coordination.

Following the reviewer's comment, we added AFM images of the cyclo-dipeptides at the lower concentrations to the SI section, expended the discussion of this point and revised the manuscript as follows:

Page 16, Line 19: Specifically, cyclo-HH formed nanofibers in MeOH, cyclo-FF formed spherical nanoparticles in DMSO, and cyclo-YY assembled into nanorods in DMSO (Fig. 7, panel *ii*). AFM analysis demonstrated small aggregates less than 4 nm in height at a lower concentration (0.5 mM), with dots for cyclo-HH, cyclo-FF and thin nanofibers for cyclo-YY (Fig. S24). This suggested that

like cyclo-FW and cyclo-WW, the self-assemblies of these three cyclo-dipeptides were also composed of QC intermediates.

-Which is the zeta potential of the nanoparticles?. This point is relevant for future applications.

Response: Because the solvents used in the work are organic solvents (methanol and DMSO), there are no electric double layers outside the self-assemblies. Thus, the zeta potential of the self-assembling peptide nanoparticles should be zero.

-In the *in vivo* and *in vitro* experiments which is the concentration of nanoparticles that have been tested?

Response: In the *in vitro* cytotoxicity experiment, the concentrations of nanoparticles used were 62.5 nM and 125 nM. In the *in vivo* bio-imaging experiment, the concentration used was 2.7 mM.

Following the reviewer's comment, we clarified this point and revised the manuscript as follows:

Page 16, Line 6: **c**, Cytotoxicity test of the cyclo-WW+Zn(II) nanospheres in DMSO (62.5 nM and 125 nM) towards B16-BL6, HaCaT and MCF7 cells. Viability relative to untreated controls \pm sd is shown. **d**, *In vivo* whole body NIR fluorescent imaging following subcutaneous injection of the nanospheres (50 μ L, 2.7 mM) into nude mice, showing notable emissions under various excitations. The dotted circle indicates the location of injection.

-The nanoparticles in which solvent have been administrated?

Response: We thank the reviewer for allowing us to clarify this experimental point. For the *in vitro* cytotoxicity assay experiment, the nanoparticles DMSO solution was diluted with cell culture medium to the desired concentration. For the *in vivo* bio-imaging experiment, the nanoparticles DMSO solution was diluted with water to the desired concentration, and then administered.

Following the reviewer's comment, we clarified this point and revised the manuscript as follows:

Page 25, Line 5: The medium was then replaced with a medium containing cyclo-WW+Zn(II) nanospheres at two different concentrations (62.5, 125.0 nM in cell culture medium).

Page 25, Line 11: ***In vivo* NIR imaging.** Cyclo-WW+Zn(II) nanospheres (50 μ L, 2.7 mM, diluted in water) were administered into nude mice by subcutaneous injection.

-Is it possible to functionalize the obtained nanoparticles for targeting? Please discuss this point.

Response: We thank the reviewer for raising this intriguing application of our work. Through covalent conjugation or co-assembly with functional moieties, peptide self-assemblies can be easily functionalized. Furthermore, the side-chain aromatic rings of short aromatic peptides can be modified with substituent groups. Therefore, these aromatic short cyclo-dipeptides nanoparticles can be easily functionalized with variable groups, including specific targeting moieties, such as cell adhesion motifs (IKAV, RGDS) (*Chem. Soc. Rev.* 2016, **45**, 3935-3953). Therefore, a series of applications can be envisioned in the future, such as tumour labelling, targeted drug delivery/release tracking, and even photo-therapy.

Following the reviewer's suggestions, we emphasized this point, added the relevant reference, and revised the manuscript as follows:

Page 15, Line 12: Notably, the fluorescent signals were stable, showing no decay for one week, thus highlighting the possibility of utilizing the photoluminescent QC assemblies for *in vivo* bio-imaging

applications. In addition, the advantage of easy modifications, such as specific targeting and controllable assembly/dis-assembly, facilitates simple functionalization of the assemblies⁵⁶, thus exemplifying their promising potential for targeted therapy and controllable drug release.

-This nanoparticles can be useful for photodynamic therapy?. By irradiation is possible to form radicals in the in vivo conditions?

Response: We thank the reviewer for pointing out another interesting implication of our work. Because the indole ring is photo-sensitive and can be easily oxidized, which can notably influence the supramolecular morphologies and photoluminescent properties (Fig. 2d, e and Fig. 3e, f), the self-assembly dynamics of the W-based cyclo-dipeptides can be optically controlled, potentially facilitating the corresponding photo-induced application.

However, the wavelength used for redox of indole ring is in the UV region (365 nm in our work), which is too short to permeate skin, in addition to being a mutagenic agent, posing a major challenge for *in vivo* photo-applications. On the other hand, *in vitro* photodynamic applications can be further explored.

Following the reviewer's comment, we emphasized this point and revised the manuscript as follows:

Page 8, Line 4: A similar emission was also observed after irradiation with UV light (365 nm) (cyclo-WW+UV), due to the UV-induced radical oxidation (Fig. 2e), implying a potential for photo-stimulated applications.

-Which is the stability of the nanoparticles in cell culture media or in vivo?

Response: Due to the lack of hydrophilic terminal groups (amino and carboxyl groups), the solubility of cyclo-peptides dramatically decreases as the polarity of the solvent increases. For example, our results showed that the self-assembled cyclo-WW+Zn(II) nanoparticles are notably larger in polar DMSO (Fig. 5a) than in MeOH (Fig. 3c). When changing the solvent to aqueous solutions, the self-assembly further progresses. In fact, the cyclo-dipeptides used in this work (cyclo-FW and cyclo-WW) crystallize in water (data not shown). Thus, the nanoparticles will further aggregate in cell culture medium or *in vivo*, with growth dynamics depending on peptide concentrations and solution conditions (ionic strength, pH...). Our investigations showed that the nanoparticles can be stable in cell culture media or *in vivo* for one week.

-Are the metal or the molecules released from the nanoparticles in the biological media?

Response: We thank the reviewer for raising this point. As we discussed in the previous comment, short aromatic cyclo-dipeptides (cyclo-FW and cyclo-WW) are more apt to self-assemble in polar solvents, such as biological media (which include high levels of salts), and actually crystallize in water. Also, cyclo-peptides are generally tolerant against enzymatic digestion. Therefore, the molecules will further aggregate, rather than be released, in biological media. In contrast, crystallographic analysis showed that metal ions did not exist in cyclo-dipeptides crystals, implying that during crystallization, metal ions were probably released.

Following the reviewer's comments, we emphasized this point and revised the manuscript as follows:

Page 15, Line 12: Notably, the fluorescent signals were stable, showing no decay for one week, thus highlighting the possibility of utilizing the photoluminescent QC assemblies for *in vivo* bio-imaging applications.

On the other hand, in the conclusion section it is mentioned an application for neuronal cells, please explain this point. Is not clear.

Response: We thank the reviewer for allowing us to further discuss the potential applications of our work. Self-assembling peptide semiconductors may connect the semiconductor field and biological systems (*Science*, 2017, **358**, eaam9756). Peptide dots, as demonstrated in this work, can act as artificial transmitters to allow implantation of the assemblies into neuronal cells in order to alter the synapses and neuronal excitability with the intention of controlling the brain's electrical impulses. Thus, recovery of the sensory functions in abnormal neuronal interfaces can be accomplished. Furthermore, the influence of physiochemical parameters on the optical, electric and electrochemical properties of the bio-inspired self-assemblies, and their unique photoluminescent ability to tune the signal transductions generated from neuronal can be explored, allowing to control and restore the sensory functions. In this scenario, with the advantages of intrinsic biocompatibility, and outstanding optical and electronic properties, the self-assembling peptide quantum confined nanostructures may provide a novel approach for neuronal diagnosis and therapy.

Following the reviewer's comments, we further elaborated this point and revised the manuscript as follows:

Page 18, Line 10: In addition, their nanoscale sizes, opto-electric properties and intrinsic biocompatibility allow these nanostructures to be implanted into neuronal cells in order to investigate the interface between the structures and neurons⁵⁹. Therefore, the metabolic procedures of the QC assemblies in neural cells and their response and influence on neuronal activities (such as current formation, synaptic activities and signal transduction) can be studied by tracking their signals. This can provide the basis for diagnosis and treatment of sensory functions.

REVIEWERS' COMMENTS:

Reviewer #1 (Remarks to the Author):

It seems that the authors have addressed all major issues in the manuscript and it can be accepted in the revised form.

Reviewer #2 (Remarks to the Author):

The revised manuscript by large addressed most of the raised queries along with stating some limitation to answer all. Overall this contribution will advance our knowledge in the field and will open further work in this direction. Thus this contribution may be published.

Reviewer #3 (Remarks to the Author):

The authors answer most of the reviews comments. However there are some points that should be addressed before publication.

- a) Please comment about the statistics of figure 6 c (cell viability assay). Specify which are the controls.
- b) Please comment about the zeta potential of the nanoparticles in aqueous solution.
- c) How were determined the concentration of nanoparticles solution?

A point-by-point response to the reviewers' comments

REVIEWERS' COMMENTS:

Reviewer #1 (Remarks to the Author):

It seems that the authors have addressed all major issues in the manuscript and it can be accepted in the revised form.

Reviewer #2 (Remarks to the Author):

The revised manuscript by large addressed most of the raised queries along with stating some limitation to answer all. Overall this contribution will advance our knowledge in the field and will open further work in this direction. Thus this contribution may be published.

Reviewer #3 (Remarks to the Author):

The authors answer most of the reviewers' comments. However there are some points that should be addressed before publication.

a) Please comment about the statistics of figure 6 c (cell viability assay). Specify which the controls are.

Response: The control groups were untreated cells. The cytotoxicity of the peptide nanoparticles was determined based on the viability of the samples treated with cyclo-WW+Zn(II) nanospheres relative to the untreated controls. Three technical replicates were performed and cell viability is given as a mean value. Error bars represent s.d. (n=3).

Following the reviewer's comment, we added this information and revised the manuscript as follows:

Page 25, Line 10: "The medium was then replaced with a medium containing cyclo-WW+Zn(II) nanospheres at two different concentrations (62.5, 125.0 nM in cell culture medium), or with naïve medium as a control."

b) Please comment about the zeta potential of the nanoparticles in aqueous solution.

Response: Due to the absence of free amino or carboxyl groups, cyclo-dipeptides generally display a lower charge than their linear counterparts. Specifically, for tryptophan-based cyclo-dipeptides, since the side-chain indole rings can be protonated/deprotonated under different pH conditions, the signals and magnification of the charges assumed by the self-assemblies are greatly dependent on the solution conditions. Therefore, at a very acidic pH, cyclo-WW self-assemblies assumed positive charges, with a measured zeta-potential of $+13.9 \pm 1.9$ mV at pH 1.0 (based on three measurements and averaged). As the pH increased, the indole rings were gradually deprotonated and the zeta-potential started to decrease. Furthermore, the hydroxyl ions could adsorb onto the surfaces, leading the nanoparticles to assume negative charges, with a measured zeta-potential of -33.5 ± 2.2 mV at pH 8.7 and -21.0 ± 1.3 mV at pH 13.0 (both based on three measurements and averaged).

c) How were determined the concentration of nanoparticles solution?

Response: Our findings revealed that the photoluminescence of peptide self-assemblies was aggregation-induced emission (Fig. 1). Therefore, through comparison of excitation wavelengths, the required peptide concentrations could be determined. For example, tryptophan amino acids have a maximal excitation at 285 nm, while after aggregation the excitation red-shifted to 305 nm (Fig. 1b). Therefore, based on the excitation wavelength, the peptide concentrations could be chosen, thereby distinguishing between the monomer versus the self-assembled state.

For cytotoxicity and *in vivo* bio-imaging experiments, the peptides self-assembly solutions with identified concentrations (larger than the critical aggregation concentration) were first prepared in DMSO, in order to ascertain the self-assembly process. A small volume of DMSO solution was then added into the required volume of cell culture medium or water for cytotoxicity and *in vivo* bio-imaging experiments, respectively. The nanoparticles concentrations were mathematically calculated according to the dilution ratios. Before the biological experiments, fluorescence characterizations were performed to validate that the nanoparticles were not disassembled after dilution.